# GraphRAG-Bench: Challenging Domain-Specific Reasoning for Evaluating Graph Retrieval-Augmented Generation

## Abstract

Graph Retrieval-Augmented Generation (GraphRAG) has garnered increasing recognition for its potential to enhance large language models (LLMs) by structurally organizing domain-specific corpora and facilitating complex reasoning. However, current evaluations of GraphRAG models predominantly rely on traditional question-answering datasets. Their limited scope in questions and evaluation metrics fails to comprehensively assess the reasoning capacity improvements enabled by GraphRAG models. To address this gap, we introduce GraphRAG-Bench, a large-scale, domain-specific benchmark designed to rigorously evaluate GraphRAG models. Our benchmark offers three key superiorities: $(i)$ Challenging question design. Featuring college-level, domain-specific questions that demand multi-hop reasoning, the benchmark ensures that simple content retrieval is insufficient for problem-solving. For example, some questions require mathematical reasoning or programming. $(ii)$ Diverse task coverage. The dataset includes a broad spectrum of reasoning tasks, multiple-choice, true/false, multi-select, open-ended, and fill-in-the-blank. It spans 16 disciplines in twenty core textbooks. $(iii)$ Holistic evaluation framework. GraphRAG-Bench provides comprehensive assessment across the GraphRAG pipeline, including graph construction, knowledge retrieval, and answer generation. Beyond final-answer correctness, it evaluates the logical coherence of the reasoning process. By applying nine contemporary GraphRAG methods to GraphRAG-Bench, we demonstrate its utility in quantifying how graph-based structuring improves model reasoning capabilities. Our analysis reveals critical insights about graph architectures, retrieval efficacy, and reasoning capabilities, offering actionable guidance for the research community.

## 1 Introduction

Retrieval-Augmented Generation (RAG) Lewis et al. (2020); Gao et al. (2024) has emerged as a key solution to ground large language models (LLMs) in external knowledge to mitigate both the hallucination problem and the lack of domain knowledge. By retrieving relevant text passages from corpora, RAG injects factual knowledge for a more reliable generation from LLMs. However, conventional RAG systems remain unsatisfactory when dealing with complex reasoning scenarios. The flat retrieval in RAG directly returns fragmentized chunks based on similarity matching, which limits their ability to model complex relationships between concepts to answer the questions requiring multi-hop reasoning Zhang et al. (2025); Dong et al. (2023), i.e., 'What was the impact of [event] the 2008 Lehman Brothers bankruptcy on [person] Elon Musk's Tesla?' or global comprehension, i.e., 'What is the main idea of the [event] Trade Policy Change?'.

To address these limitations, Graph Retrieval-Augmented Generation (GraphRAG) has been extensively studied to capture the structured knowledge among concepts in the form of graphs Edge et al. (2025); Peng et al. (2024); Zhou et al. (2025), where nodes represent concepts and edges are for the relations among them. Recent advances in GraphRAG can be categorized into three main directions. First, hierarchical graph construction methods like RAPTOR Sarthi et al. (2024) and Microsoft's GraphRAG Edge et al. (2025) organize knowledge through tree structures and community detection. Second, neural graph retrieval approaches, including GFM-RAG Luo et al. (2025) and G-Retriever He et al. (2024) employ graph neural encoders with specialized objectives for multi-

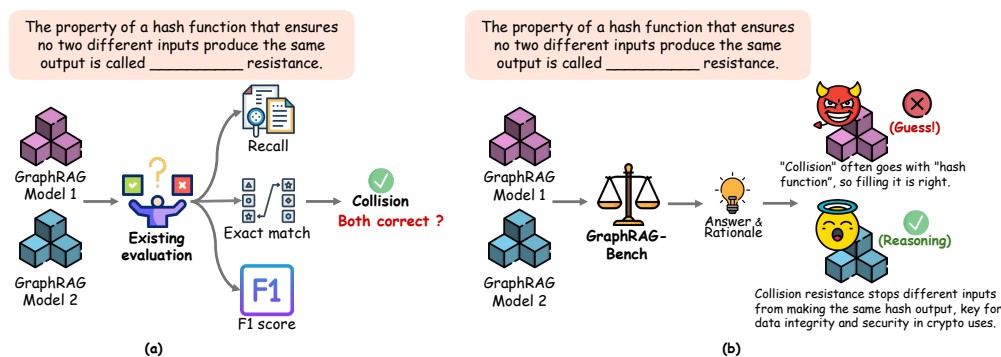

Figure 1: Comparison between existing evaluations (a) and our proposed GraphRAG-Bench (b). GraphRAG-Bench not only assesses the accuracy of generation but also evaluates the rationality of reasoning based on the challenging domain-specific questions.

hop reasoning. Third, dynamic knowledge integration systems such as DALK Li et al. (2024) and ToG Sun et al. (2024) develop adaptive graph construction and traversal mechanisms that are tightly coupled with LLMs. By structuring knowledge as graphs, GraphRAG enables LLMs to both traverse and reason over explicit relational paths, but also supports deeper reasoning by inferring implicit relations based on the graph structure Dong et al. (2025).

However, despite the promise, existing benchmarks for GraphRAG methods fail to reflect the performance of reasoning on graphs. They predominantly leverage the traditional QA dataset, e.g., HotpotQA Yang et al. (2018), 2WikiMultiHopQA Ho et al. (2020) and MuSiQue Trivedi et al. (2022), which only feature explicit factoid questions with limited complexity and short answers, e.g., 'Who is the grandchild of Dambar Shah?'. These datasets suffer from three critical limitations: $(i)$ There are only commonsense questions that could be probably covered in the training corpus of LLMs. $(ii)$ They typically require only single-hop or shallow multi-hop reasoning based on explicit connections, which inadequately probes the unique advantages of graph-structured knowledge. $(iii)$ Narrow Answer Formats. Most answers are short (names, dates) or multiple-choice, which could hardly reflect the reasoning ability over graphs. To this end, we would like to ask a research question:

***"Does graph augmentation truly enhance reasoning capabilities beyond simple retrieval?"***

In this paper, we propose GraphRAG-Bench, the first challenging domain-specific benchmark particularly designed for GraphRAG. $(i)$ Our dataset contains 1,018 college-level question spans 16 disciplines, e.g., computer vision, networks, human-computer interaction, etc, featuring the ability of conceptual understanding, e.g., "Given [theorem] A and B, prove [conclusion] C", complex algorithmic programming, e.g., coding with interlinked function calls) and mathematical computation, e.g., "Given [Input], [Conv1], [MaxPool], [FC], calculate the output volume dimensions." $(ii)$ GraphRAG-Bench contains five types of diverse questions to thoroughly evaluate different aspects of reasoning, including multiple-choice (MC), multi-select (MS), true-or-false (TF), fill-in-blank (FB) and open-ended (OE). $(iii)$ We offer a comprehensive multi-dimensional evaluation on each component of GraphRAG, including graph construction, knowledge retrieval, answer generation and rationale generation. We aim to provide unprecedented insights into how graph-structured knowledge enhances LLMs' reasoning capabilities compared to traditional RAG approaches. The major contributions are summarized hereunder:

- We propose the first challenging domain-specific benchmark, particularly concentrating on GraphRAG. It contains 1018 questions in 5 question types spanning 16 topics and a corpus of 7 million words from 20 computer science textbooks.

- A comprehensive evaluation protocol is designed to stress-test GraphRAG methods on graph construction, retrieval, and multi-hop answer generation and rationale generation.

- Extensive experiments have been conducted with nine state-of-the-art GraphRAG models. We make insightful observations and provide the insights that: 1) GraphRAG substantially enhances the reasoning capabilities of LLMs, and - to the best of our knowledge - we are the first to quantify this improvement using concrete evaluation metrics. 2) GraphRAG's impact varies by question types: it yields significant gains on some types but offers limited benefit for others.

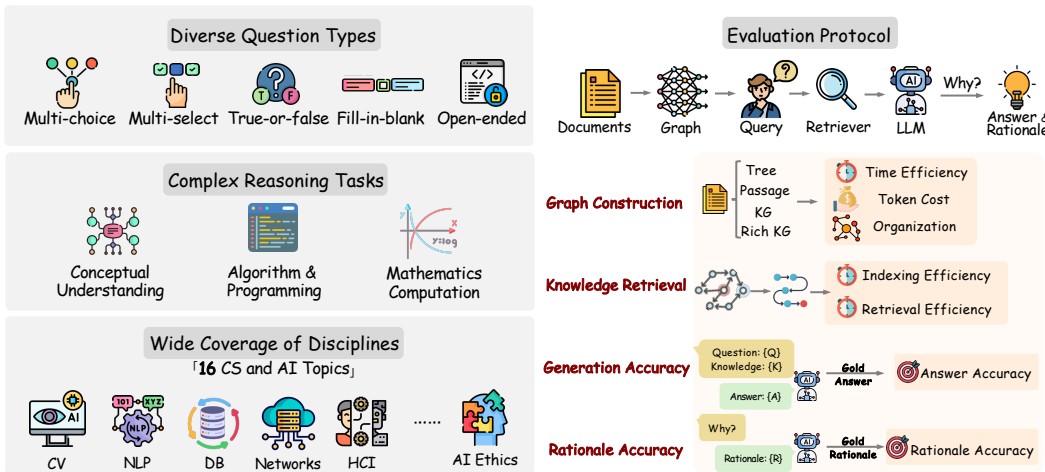

Figure 2: The overview of our benchmark `GraphRAG-Bench`, illustrating the contributions.

## 2 GRAPHRAG-BENCH: CHALLENGING REASONING BENCHMARK

### 2.1 QUESTION DESIGN

To evaluate the GraphRAG framework on college-level reasoning, we first assembled an authoritative textbook corpus. Beginning with over 100 publications spanning 16 distinct subfields in computer science, we systematically identified the most representative 20 textbooks (the details of licenses can be found in ethics statement). We defined five types of questions, each targeting a different aspect of GraphRAG's reasoning capabilities, which are detailed in Tab. 1. After rigorous screening and refinement by several domain experts, we selected 1,018 high-quality challenging questions, covering a broad spectrum of topics. The details of question design and selection can be found in the Appendix.

By design, each question type is explicitly mapped to the core competencies of GraphRAG, with individual questions meticulously crafted for application in college-level instructional or assessment contexts. Should GraphRAG demonstrate improved performance on these tasks, it would establish itself as a highly effective tool in education, significantly enhancing teaching and learning efficiency.

Table 1: The description of different question types.

| Question Type | Description |
| --- | --- |
| Fill-in-blank (FB) | Requires completing context-dependent statements with semantically precise terms. These assess the model's ability to generate coherent content by leveraging local semantic dependencies and entity grounding within graph-structured knowledge. |
| Multi-choice (MC) | Presents a question with 4 options, including linguistically plausible distractors. These assess the model's capacity to discern correct answers through discriminative reasoning, integrating entity information and edge relationships to reject semantically similar but factually incorrect options. |
| Multi-select (MS) | Demands selecting 2–4 correct answers from 4 options, often requiring reasoning over interconnected concepts. The inclusion of overlapping distractors tests the model's ability to handle complex query semantics, aggregating evidence from multi-hop graph paths and resolving conflicts between related but non-essential attributes. |
| True-or-false (TF) | Involves verifying the correctness of statements. These measure the model's factual accuracy, requiring logical inference over knowledge. |
| Open-ended (OE) | OE questions allow for a wide range of responses, requiring methods to formulate detailed and comprehensive answers. These evaluate the model's holistic knowledge synthesis, demanding the integration of multi-subfield knowledge to generate structured, logically coherent long-form responses. |

## 2.2 Corpus collection and processing

Extracting accurate content from 20 PDF textbooks is challenging. We implement a multi-stage pipeline comprising preprocessing, content parsing, post-processing, and hierarchy construction, the details of which can be found in the Appendix. In preprocessing we separate text and scanned pages by text density and image-area ratio, extracting text with PyMuPDF or OCR as needed, and gather metadata (outline, page count, chapter/section ranges). For parsing we apply LayoutLMv3 Huang et al. (2022) to segment semantic regions (titles, paragraphs, figures, tables), detect formulas with a YOLO-based model Wang et al. (2024a) so formula images are handled separately, and transcribe scanned regions with PaddleOCR in reading order. Post-processing merges and reorders fragmented or overlapping regions into natural reading order using MinerU Wang et al. (2024b).

Finally, we organize the extracted content into a hierarchical textbook-tree structure. We map the textbook metadata (e.g., chapter titles, section divisions, and page ranges) to a four-level hierarchy: Book Title → Chapter → Section (Subchapter) → Knowledge Content Unit. Each node in this hierarchy is annotated with its contextual metadata and its structural role. This textbook-tree provides an intuitive navigation framework aligned with the textbook's organization. The resulting corpus, with its accurate content extraction, structural annotation, and hierarchical organization, forms a robust basis for evaluating GraphRAG's ability to leverage organized textbook knowledge for context-rich reasoning and retrieval-augmented generation.

## 2.3 Expert-crafted rationale

Existing benchmarks typically supply only final answers or explicit graph paths; by contrast, our dataset supplies expert-crafted rationales that articulate the complete logical progression necessary to solve each problem. These rationales go beyond mere corpus aggregation; they are structured narratives that (i) isolate prerequisite concepts, (ii) describe the relationships among these concepts, and (iii) specify the inferential operations applied during problem solving. By tracing each step of logical inference and knowledge interaction, we can assess whether GraphRAG models truly generate contextually grounded explanations or simply exploit surface-level patterns.

To enable fine-grained, topic-specific evaluation, each question in our dataset carries two hierarchical labels: a broad subfield (Level 1, e.g., "Machine Learning") and a more granular concept (Level 2, e.g., "Unsupervised Learning"). These annotations structure our post-hoc analyses. For each topic, we measure not only the accuracy of the model's answer but also the degree to which its generated rationale aligns with the gold one. In this way, we convert evaluation into a multidimensional process, requiring models to produce both correct solutions and faithful reasoning patterns.

## 3 Experiments

We conduct experiments on each submodule following GraphRAG's pipeline, which includes the **graph construction** (or similar specialized structures), **knowledge retrieval**, and **generation**. Additionally, since our dataset contains a gold rationale for each query, we require the GraphRAG method to generate **rationales** during the generation phase to evaluate its reasoning capabilities.

**Metrics.** We provide a succinct introduction to the core ideas of each metric; the full evaluation protocol and details can be found in the Appendix.

- **Graph construction.** We evaluate graph construction across three aspects: 1) Efficiency: the time required to build a complete graph. 2) Cost: the number of tokens consumed during graph construction. 3) Organization: the proportion of non-isolated nodes within the constructed graph.

- **Knowledge retrieval.** We evaluate retrieval from two dimensions: 1) indexing time, defined as the duration required to construct the vector database for retrieval; 2) average retrieval time, representing the mean time consumed for retrieval per query. Additionally, we summarize the retrieval operators employed by each method to assess the complexity of their retrieval mechanisms.

- **Generation.** We argue that the existing exact match metric is inappropriate, as correct answering does not necessitate word-by-word correspondence. Therefore, this paper introduces a new metric, Accuracy, defined as follows: 1) For OE and FB questions, both the output and groundtruth are fed into an LLM via our designed prompt, which assigns a score based on semantic alignment and

correctness. 2) For MC and TF, 1 point for the correct answer, 0 points for otherwise. 3) For MS, 1 point for a fully correct answer; 0.5 points for a subset; 0 points for incorrect answers.

- **Rationale.** We designed a prompt to feed both the rationale generated by GraphRAG method and gold rationale into a LLM, which assigns a reasoning score R to evaluate their semantic correspondence and reasoning consistency. Simultaneously, we developed an additional assessment metric, namely the AR metric, to determine whether the model is able to provide correct reasoning when it answers the question accurately. This metric serves to distinguish whether the model has merely guessed the correct answer or has actually engaged in proper logical reasoning to reach the correct answer, thereby offering a more comprehensive understanding of the model's performance.

**Experiment setups.** In our experiments, we evaluated the performance of nine state-of-the-art GraphRAG methods, including: 1) RAPTOR Sarthi et al. (2024); 2) LightRAG Guo et al. (2024); 3) GraphRAG Edge et al. (2025); 4) G-Retriever He et al. (2024); 5) HippoRAG Gutiérrez et al. (2024); 6) GFM-RAG Luo et al. (2025); 7) DALK Li et al. (2024); 8) KGP Wang et al. (2024c); 9) ToG Sun et al. (2024). To ensure a fair comparison across all methods, we adopted the same GPT-4o-mini as the default large language model. We imposed no max token length to limit the performance of individual methods. For methods requiring top-k selection, we uniformly set k=5. Regarding text chunking, the chunk size was consistently set to 1200 tokens. Except for the parameters standardized for fair comparison, all other parameters were configured to the optimal values reported in the original papers.

Table 2: Comparison of graph construction process.

| Category | Method | Token cost of graph construction | Time cost of graph construction | Organization |
|---|---|---|---|---|
| Tree | RAPTOR (2024) | 10,142,221 | 20396.49s | - |
| Passage Graph | KGP (2024) | 15,271,633 | 17318.07s | 46.03% |
| Rich KG | LightRAG (2024) | 83,909,073 | 12976.22s | 69.71% |
| Rich KG | GraphRAG (2025) | 79,929,698 | 11181.24s | 72.51% |
| KG | G-Retriever (2024) | 32,948,161 | 5315.27s | 89.95% |
| KG | HippoRAG (2024) | 33,006,198 | 5051.41s | 89.58% |
| KG | DALK (2024) | 33,007,324 | **4674.30s** | 89.49% |
| KG | ToG (2024) | 33,008,230 | 5235.30s | 89.95% |
| KG | GFM-RAG (2025) | 32,766,094 | 5631.10s | **89.97%** |

### 3.1 EVALUATION OF GRAPH CONSTRUCTION

Graph construction aims to transform corpus into structured, storable objects, serving as the foundational step in GraphRAG. Current mainstream graph construction methods can be categorized into four classes: 1) Tree: RAPTOR leverages this structure, where each leaf node represents a chunk. By generating summaries via LLMs and applying clustering methods, parent nodes are iteratively created to form a tree structure. 2) Passage Graph: Adopted by KGP, this structure represents each chunk as a node, with edges established through entity linking tools. 3) Knowledge Graph: Used in G-Retriever, HippoRAG, GFM-RAG, and DALK, this structure extracts entities and relationships from chunks using open information extraction (OpenIE) tools to construct knowledge graphs. 4) Rich Knowledge Graph: Employed by GraphRAG and LightRAG, this structure enriches standard knowledge graphs with additional information (e.g., summarizing descriptions for nodes or edges).

Experimental results in Tab. 2 show that the tree structure incurs the lowest token count, as it only invokes LLMs for summary generation, but requires the longest time due to iterative clustering. The passage graph has suboptimal token cost, invoking LLMs only for summarizing entities or relationships, with the second-longest time consumption attributed to the time-intensive entity linking process. The knowledge graph has moderate token usage, requiring LLMs for both entity extraction from corpora and triple generation from entities, yet achieves the shortest time consumption due to rapid knowledge graph construction after triple acquisition. The rich knowledge graph consumes the most tokens, as it generates additional descriptions for entities and relationships via LLMs on top of standard knowledge graphs, leading to increased time costs. For evaluating graph construction quality, we use the non-isolated nodes ratio as the metric. Since the Tree structure contains no isolated nodes, this metric is inapplicable to it. Experimental results show that the Knowledge Graph achieves the best performance, with its non-isolated nodes ratio maintained at approximately 90%.

The Rich Knowledge Graph performs suboptimally; while it incorporates additional information, it inevitably introduces more noise. The Passage Graph exhibits the lowest non-isolated nodes ratio, indicating that entity linking tools fail to effectively establish edges between most entity pairs.

Table 3: Comparison of knowledge retrieval process.

| Category | Method | Retrieval operators | Indexing time | Average retrieval time |
|---|---|---|---|---|
| Passage Graph | KGP | Node | 204.10s | 89.38s |
| Rich KG | GraphRAG | Node+Relationship+Chunk+Community | 1796.65s | 44.87s |
| Rich KG | LightRAG | Node+Relationship+Chunk | 1430.32s | 13.95s |
| KG | ToG | Node+Relationship | 1080.43s | 70.53s |
| KG | DALK | Node+Subgraph | 407.10s | 26.80s |
| KG | G-Retriever | Node+Relationship+Subgraph | 920.39s | 23.77s |
| KG | HippoRAG | Node+Relationship+Chunk | 4695.29s | 2.44s |
| KG | GFM-RAG | Node | **93.55s** | 1.96s |
| Tree | RAPTOR | Node | 451.03s | **0.02s** |

## 3.2 EVALUATION OF KNOWLEDGE RETRIEVAL

As shown in Tab. 3. GFM-RAG incurs the shortest indexing time; it does not construct a traditional vector database to store entities but instead stores question-corresponding entities exclusively during graph construction. Among methods using vector databases, KGP, RAPTOR, and DALK exhibit lower costs due to minimal stored information; ToG, G-Retriever, and LightRAG have moderate costs, as relationship storage is inherently time-consuming; GraphRAG further increases indexing time by additionally storing community reports. HippoRAG demands the longest indexing time, attributed to its extra construction of entity↔relationship and relationship↔chunk mappings. Regarding average retrieval time, RAPTOR achieves the fastest speed, as its tree structure enables rapid information localization. GFM-RAG and HippoRAG follow, leveraging GNNs and PageRank for retrieval, respectively. G-retriever employs a prize-collecting Steiner forest algorithm, while LightRAG relies on relationship-based retrieval, both introducing additional latency. GraphRAG needs to utilize community information for retrieval, which leads to its time-consuming. KGP, ToG, and DALK incur substantial time costs due to their dependence on LLM invocations during retrieval.

Table 4: Comparison of generation process.

| Category | Method | Accuracy | | | | | |
|---|---|---|---|---|---|---|---|
| | | Fill-in-blank | Multi-choice | Multi-select | True-or-false | Open-ended | Average |
| Base LLM | GPT-4o-mini | 74.29 | 81.11 | 76.68 | 75.95 | 52.23 | 70.68 |
| Naive RAG | TF-IDF | 75.71 | 77.88 | 72.52 | 84.17 | 50.18 | 71.71↑ |
| | BM-25 | 74.28 | 78.80 | 71.17 | **84.49** | 50.00 | 71.66↑ |
| | BERT-large | 71.43 | 79.26 | 74.77 | 81.33 | 51.86 | 71.32↑ |
| | BGE-M3 | 77.62 | 77.42 | 68.02 | 82.60 | 53.35 | 71.66↑ |
| Passage Graph | KGP | 74.29 | 79.26 | 74.77 | 82.28 | 51.49 | 71.86↑ |
| Rich KG | LightRAG | 65.24 | 78.80 | 73.42 | 82.59 | 53.16 | 71.22↑ |
| Rich KG | GraphRAG | 75.24 | **81.57** | 77.48 | 80.70 | 52.42 | 72.50↑ |
| KG | DALK | 70.00 | 78.34 | 71.62 | 77.22 | 51.49 | 69.30↓ |
| KG | G-Retriever | 70.95 | 77.42 | 71.62 | 78.80 | 52.04 | 69.84↓ |
| KG | ToG | 70.48 | 78.80 | **78.38** | 79.75 | 54.28 | 71.71↑ |
| KG | GFM-RAG | 72.38 | 80.65 | 72.07 | 82.59 | 52.79 | 72.10↑ |
| KG | HippoRAG | 70.48 | 80.18 | 74.32 | 81.65 | **56.13** | 72.64↑ |
| Tree | RAPTOR | **76.67** | 80.65 | 77.48 | 82.28 | 54.83 | **73.58**↑ |

## 3.3 EVALUATION OF GENERATION ACCURAY

As shown in Tab.4. Given that GPT-4o-mini already exhibits strong question-answering capabilities, not all GraphRAG methods effectively enhance its performance. Notably, DALK and G-Retriever degrade LLM performance; their over-reliance on structural information at the expense of semantic

content introduces excessive noise during generation, impairing LLM judgment accuracy. LightRAG, ToG, and KGP achieve slight performance improvements, indicating their retrieved content provides marginal assistance for generation tasks. In contrast, GFM-RAG, GraphRAG, and HippoRAG significantly boost LLM performance by effectively integrating graph structural information with chunk-level semantics: GFM-RAG leverages large-scale pretraining to obtain a robust foundation model, GraphRAG optimizes retrieval using community-based information, and HippoRAG enhances retrieval efficiency via PageRank algorithm. The top-performing method in experiments is RAPTOR, which constructs a tree structure through iterative clustering, a design that aligns with the natural hierarchical organization of textbook data, enabling efficient retrieval of relevant information. Additionally, most GraphRAG methods outperform traditional RAG baselines such as BM-25 and TF-IDF, highlighting the utility of graph-based architectures in improving generation accuray.

Table 5: Comparison of reasoning capability.

| Category | Method | Reasoning | | | | | | | | | | | |
| | | FB | | MC | | MS | | TF | | OE | | Average | |
| | | R | AR | R | AR | R | AR | R | AR | R | AR | R | AR |
| Base LLM | GPT-4o-mini | 64.76 | 53.33 | 55.07 | 50.92 | 54.50 | 39.19 | 58.23 | 53.40 | 49.26 | 9.76 | 55.45 | 39.78 |
| Naive RAG | TF-IDF | 68.09 | 52.61 | 52.76 | 49.19 | 56.30 | 43.02 | 64.08 | 61.23 | 50.37 | 10.50 | 57.61 | 42.38 |
| | BM-25 | 69.04 | 56.42 | 57.14 | 53.11 | 57.20 | 42.79 | 65.18 | 62.18 | 50.74 | 11.52 | 59.18 | 44.15 |
| | BERT-large | 67.62 | 52.14 | 56.22 | 52.07 | 55.41 | 44.59 | 65.66 | 62.26 | 52.60 | 10.97 | 58.99 | 43.27 |
| | BGE-M3 | 67.14 | 53.10 | 54.61 | 51.04 | 58.11 | 44.14 | 63.77 | 60.36 | 52.97 | 11.34 | 59.01 | 43.37 |
| Passage Graph | KGP | 64.29 | 49.29 | 56.45 | 52.07 | 58.11 | 44.37 | 64.08 | 60.68 | 52.42 | 8.92 | 58.74 | 42.22 |
| Rich KG | GraphRAG | **71.43** | 55.24 | 56.22 | 52.42 | 57.66 | 45.72 | 63.61 | 60.13 | 53.16 | 10.50 | 59.43 | 43.30 |
| Rich KG | LightRAG | 66.19 | 47.86 | 57.14 | 52.30 | **61.71** | **49.10** | 66.61 | 63.45 | 53.16 | 10.13 | 60.46 | 43.81 |
| KG | DALK | 70.95 | 55.24 | 54.15 | 50.35 | 59.01 | 46.40 | 62.18 | 58.23 | 54.09 | 9.67 | 58.89 | 42.12 |
| KG | G-Retriever | 70.00 | 55.00 | **57.60** | **53.46** | 60.81 | 48.20 | 64.24 | 60.21 | 53.35 | 10.04 | 60.17 | 43.66 |
| KG | ToG | 70.00 | 53.10 | 56.00 | 51.73 | 57.21 | 45.72 | 65.66 | 62.26 | 54.46 | 12.08 | 60.17 | 44.01 |
| KG | GFM-RAG | 70.00 | 54.76 | 56.22 | 52.07 | 58.11 | 45.50 | 66.46 | **63.69** | 53.72 | 10.69 | 60.36 | 44.30 |
| KG | HippoRAG | 66.67 | 50.48 | 56.68 | 52.30 | 59.91 | 47.52 | **67.25** | 63.61 | **55.02** | 12.36 | **60.90** | 44.55 |
| Tree | RAPTOR | **71.43** | **57.86** | 56.45 | 52.07 | 60.36 | **49.10** | 66.30 | 62.90 | 53.90 | **13.57** | 60.81 | **45.53** |

## 3.4 EVALUATION OF REASONING CAPABILITY

As shown in Tab.5. In contrast to the high accuracy in generation tasks, GPT-4o-mini exhibits a notable decline in reasoning performance. The decrease in R score indicates that LLMs often fail to perform correct reasoning, instead selecting answers through conjecture or pattern matching in many cases. The drop in AR score suggests that even when LLMs provide correct answers, their reasoning processes may be flawed; alternatively, they might generate correct reasoning but choose incorrect answers. Importantly, all GraphRAG methods significantly enhance the reasoning capabilities of LLMs: through distinct designs, these methods retrieve not only semantically relevant corpus for questions but also identify multi-hop dependent corpus in the knowledge base, providing evidential support for LLM reasoning. This enables LLMs to reason based on external information rather than relying solely on internal knowledge for conjecture. In terms of algorithm performance, the distribution aligns with that of generation tasks: HippoRAG and RAPTOR remain the top performers, which is intuitive, since retrieving useful information is inherently correlated with enabling correct reasoning. Additionally, most GraphRAG methods still outperform traditional RAG baselines.

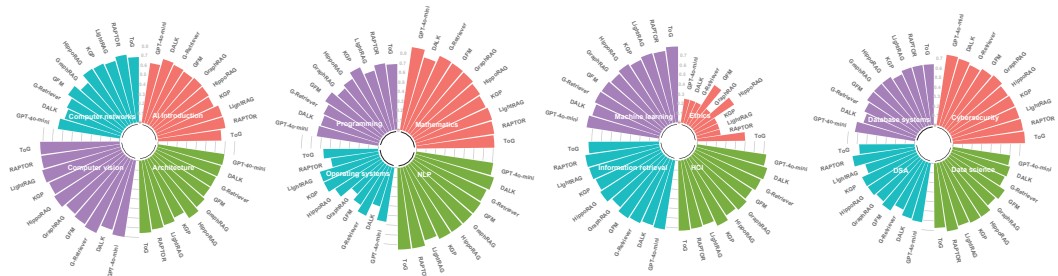

Figure 3: Comparison of Generation Accuracy by Topic.

### 3.5 TOPIC-SPECIFIC GENERATION ACCURACY ANALYSIS

Given our dataset spans 16 distinct topical domains, we conducted a fine-grained analysis of GraphRAG's impact on LLM generation accuracy. Overall, GraphRAG yields consistent improvements in most areas; However, several intriguing findings emerge: **1) Mathematics Domain.** All GraphRAG methods degrade the LLM's generation accuracy in mathematics. This is attributed to the critical reliance of mathematical problems on rigorous symbolic manipulation and precise reasoning chains; models must internally "compute" each deductive step rather than relying on keyword matching from external texts. Most documents retrieved through GraphRAG are explanatory or conceptual, with symbolic notation, formula layouts, and contextual structures often misaligned with the problem requirements, leading to ambiguities or loss of key steps during the extraction and transformation of information. **2) Ethics Domain.** Both GraphRAG and the LLM itself exhibit mediocre performance in ethics. We posit that ethical problems fundamentally involve subjective value judgments, whose meanings depend on dynamic contexts of moral trade-offs and social norms. The symbolic representations captured by LLMs through statistical learning struggle to accurately model ambiguous ethical constructs, introducing intrinsic limitations in reasoning. **3) Robustness.** Excellent GraphRAG approaches such as RAPTOR enhance LLM generation accuracy across most topics, demonstrating robust performance that validates their cross-domain effectiveness.

### 3.6 COMPARISON ON DIFFERENT SCALES OF CORPORA

Given that corpus sizes can vary significantly in real-world scenarios, we conducted additional experiments to further verify the effectiveness of different GraphRAG methods across corpora of varying scales. Specifically, we extracted two subsets from GraphRAG-Bench: a medium-sized subset (comprising 3–4 textbooks and their corresponding questions) and a small-sized subset (consisting of 1 textbook and its corresponding questions). Experimental results are presented in Tab. 6. Our findings indicate that as corpus size decreases, performance discrepancies between different methods become more pronounced. Nevertheless, more advanced GraphRAG methods (e.g., HippoRAG, GFM-RAG, RAPTOR) consistently retain substantial performance advantages. This further demonstrates that GraphRAG-Bench can effectively evaluate the performance of GraphRAG methods across different scales.

Table 6: Comparison on different scales of corpora

| Category | Method | Medium-sized corpora | | | Smaller-sized corpora | | |
|---|---|---|---|---|---|---|---|
| | | Accuracy | R score | AR score | Accuracy | R score | AR score |
| Passage Graph | KGP | 68.49 | 57.56 | 38.45 | 68.18 | 48.48 | 35.61 |
| Rich KG | GraphRAG | 60.92 | 54.62 | 35.08 | 65.15 | 47.73 | 29.55 |
| Rich KG | LightRAG | 67.65 | 57.14 | 38.87 | 68.94 | 46.97 | 32.58 |
| KG | DALK | 60.92 | 54.62 | 34.87 | 62.88 | 46.21 | 29.17 |
| KG | G-Retriever | 63.03 | 56.30 | 36.34 | 61.36 | 47.72 | 31.82 |
| KG | ToG | 68.07 | 56.72 | 38.45 | 67.42 | 50.00 | 34.47 |
| KG | GFM-RAG | 69.33 | 55.88 | 39.92 | 70.45 | **51.51** | 37.12 |
| KG | HippoRAG | 70.17 | **59.24** | **41.81** | 70.45 | 50.76 | 35.98 |
| Tree | RAPTOR | **71.43** | 58.40 | 41.18 | **71.97** | 50.76 | **38.26** |

### 3.7 EXPERT-GUIDED POST VERIFICATION

For Open-ended and Fill-in-blank question types and reasoning, we believed that directly using string exact match was unreasonable. Many correct statements were judged incorrectly due to descriptions, capitalization, abbreviations. In such cases, we included LLM for auxiliary judgment, which is a widely adopted paradigm in generative tasks. During the actual post-processing, to verify the accuracy of LLM-as-a-judge, we have conducted expert-

Table 7: Post verification results

| Reliability | Counts | Percentage |
|---|---|---|
| 3/3 | 470 | 94.0% |
| 2/3 | 28 | 5.6% |
| 1/3 | 1 | 0.2% |
| 0/3 | 1 | 0.2% |

guided post verification. We sampled 500 questions from the dataset and asked 3 human experts (researchers with a computer science doctoral degree) to score them. The experts judged whether the LLM's judgment was reasonable based on the LLM's generation and the groundtruth and gave 1 score if yes. 3/3 points indicated reasonable and an aligned answer, 2/3 points indicated basically reasonable since the majority of experts agreed, and 1/3 or 0/3 point indicated unreasonable and needed to be reevaluted by the experts. The specific experimental results are shown in the Tab. 7, proving the reliability of LLM-as-a-judge.

## 4 OBSERVATION

### *'Can GraphRAG improve performance across all question types?'*

**Accuracy drop of MC questions.** LLMs have internalized vast amounts of knowledge through extensive training on large corpora, enabling them to often correctly select answers in multiple-choice tasks. However, GraphRAG's retrieval-based augmentation may introduce redundant or loosely related information that does not precisely match the question context. Such retrieval noise can interfere with the model's decision-making ability, ultimately reducing its accuracy on MC questions.

**Improvement in TF questions.** TF questions require binary judgments about factual or logical statements. LLMs may contain blind spots or incomplete knowledge for certain facts, leading to incorrect answers. By retrieving relevant factual evidence, GraphRAG helps the model verify statements before answering. These supplementals improve the model's accuracy on TF questions.

**Improvement in OE questions.** Open-ended questions allow for expansive, detailed responses, which can be challenging for LLMs that rely solely on their internal knowledge. GraphRAG mitigates this challenge by providing additional context and facts from external corpora. The retrieved information enriches the model's responses, improves subject-matter detail and expressiveness, and reduces instances of hallucination by grounding answers in explicit evidence.

**Different effects in FB & MS questions.** Fill-in-blank questions demand precise contextual understanding to correctly predict missing words. GraphRAG's retrieved corpora often fail to match exact contexts, introducing noise that degrades the model's performance on FB questions. Multi-select questions require choosing multiple correct answers from a set and involve reasoning over complex combinations of options; if GraphRAG's retrieval omits relevant answer options or includes irrelevant details, it can confuse the model. As a result, these question types place high demands on retrieval precision; GraphRAG may have limited benefit unless its retrieval is highly accurate.

### *'Can GraphRAG effectively enhance LLMs' reasoning ability?'*

Experiments demonstrate that GraphRAG enhances the reasoning capabilities of LLMs across diverse question types, increasing the probability of generating correct rationales alongside answers. This is attributed to their efficient retrieval mechanisms, which not only identify relevant corpora for questions but also provide robust evidential support for LLM reasoning processes. In particular, existing benchmarks lack systematic evaluation of GraphRAG's reasoning capabilities, an aspect of critical importance in real-world applications. For example, in the college-level educational context targeted in this document, users seeking professional knowledge expect not only correct answers, but also explicit rationales to facilitate understanding and knowledge acquisition. Similarly, in medical scenarios, patients require clear rationales for medication along with treatment recommendations to ensure transparency in decision-making. Thus, an effective GraphRAG approach should aim not only for high accuracy in answer generation but also for strong reasoning and explainability. Providing clear evidence-based justifications and reasoning chains is essential for meeting the requirements of explainability and transparency in real-world scenarios.

Regarding the analysis and observation we conducted: Firstly, we obtained the phenomena through the experimental results. Secondly, the conclusions were derived by analyzing multiple instances in the specific QA process. The specific case study can be found in the Appendix.

## 5 CASE STUDY

As illustrated in Fig 4, we present a case study highlighting specific challenges within our dataset. Our questions span 16 core topics in undergraduate computer science; here, we focus on a sample

---

**Case Study**

**Question:** Why is it necessary for the server to use a special initial sequence number (ISN) in the SYN-ACK?

**Multi-hop Reasoning:**
1. <Server>→*sends*→ <SYN-ACK Packet> →*includes*→ < ISN > →*used to ensure*→ <Unique connection identification>
2. <Server>→*sends*→ <SYN-ACK Packet> →*includes*→ < ISN > →*used to ensure*→ < Proper packet sequencing >
3. < Special ISN > →*helps defend against*→ <SYN flood attack> →*exploits*→ <Predictability of ISN>

**Rationale:**
The server uses a special initial sequence number in the SYN-ACK to ensure unique connection identification and proper packet sequencing. This also mitigates SYN flood attacks by making it harder for attackers to predict ISNs and hijack sessions.

Figure 4: A case study in the topic of computer networks.

from the Computer Networks section. This example demonstrates that (i) the questions demand specialized, college-level knowledge, and (ii) the correct answer cannot be retrieved through simple lookup. Instead, solving the problem requires synthesizing multiple reasoning steps to construct a coherent rationale before generating the final answer.

# 6 CONCLUSION

In this paper, we present GraphRAG-Bench, the first domain-specific benchmark designed for GraphRAG, comprising a 16-discipline dataset that challenges methods with multi-hop reasoning, complex algorithmic/programming tasks, mathematical computing, and varied question types. Our comprehensive, multi-dimensional evaluation, spanning graph construction, knowledge retrieval, generation and reasoning, quantifies the enhancement of LLM reasoning when augmented with structured knowledge. Extensive experiments on nine state-of-the-art GraphRAG methods reveal the significant role of graph integration in improving reasoning and generation performance. Our analysis reveals critical insights about graph architectures, retrieval efficacy, and reasoning capabilities, offering actionable guidance for the research community.

ETHICS STATEMENT

**License Information**: As a pure research paper with no commercial purposes, we have already obtained full licenses to do research for 17 textbooks through the university platform among the 20 textbooks we adopted in total. For the remaining three textbooks, two of them are totally free with no copyright and one supports research with full textual resources online. For the processing of all the textbooks, we only keep the main content in the form of chunks with no tampering.

Our benchmark aims to fairly evaluate the existing open-sourced methods with the textbooks, which we have obtained the licenses. All data usage strictly adheres to the terms of the respective licenses and is confined to non-commercial research purposes. The processed textbook chunks are used solely for generating evaluation queries and will not be redistributed. We are committed to the responsible use of copyrighted materials and believe this work aligns with ethical research practices by promoting reproducible and fair comparisons.

REPRODUCIBILITY STATEMENT

To ensure complete reproducibility of our work, we have elaborately detailed the construction, parameters, and evaluation metrics of our benchmark in the main text, including key implementation details. The evaluated baselines are all open-sourced. All relevant resources covering the complete dataset and well-documented source codes have been made publicly available and can be found in the supplementary materials.

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

# A APPENDIX

## A.1 THE USAGE OF LLMS

The use of LLM in this manuscript is fully in accordance with the regulations of the ICLR, and it is only used for detecting grammatical errors.

## A.2 RELATED WORK

**GraphRAG.** Recent work in GraphRAG has focused on integrating structured knowledge and advanced retrieval strategies to overcome the limitations of vanilla RAG in handling large, noisy corpora and complex reasoning. For example, RAPTOR Sarthi et al. (2024) and Microsoft's GraphRAG Edge et al. (2025) both employ hierarchical clustering, RAPTOR via recursive tree construction with multi-level summarization, and GraphRAG via community detection with LLM-generated synopses, to support coarse-to-fine retrieval and diverse, high-coverage responses. GFM-RAG Luo et al. (2025), G-Retriever He et al. (2024), and LightRAG Guo et al. (2024) each combine graph neural encoders with specialized retrieval objectives, respectively a query dependent GNN trained in two stages for multi-hop generalizability, a Prize Collecting Steiner Tree formulation to reduce hallucination and improve scalability, and a dual level graph augmented index for efficient, incrementally updatable lookup, to enable accurate, scalable reasoning over document graphs. Inspired by hippocampal memory processes, HippoRAG Gutiérrez et al. (2024) leverages Personalized PageRank to achieve single-step multi-hop retrieval, delivering state-of-the-art efficiency and performance on both path following and path finding QA tasks. DALK Li et al. (2024) and KGP Wang et al. (2024c) introduce dynamic KG construction and traversal agents, using LLMs to build domain specific graphs and self aware retrieval policies, to inject structural context while reducing noise. ToG Sun et al. (2024) tightly couples LLMs with KGs via beam search exploration, enabling iterative graph reasoning and on the fly correction without additional training. Collectively, these methods exemplify the GraphRAG paradigm by uniting graph structures, generative language models, and novel retrieval formulations to enhance knowledge integration, scalability, and deep reasoning across diverse domains.

**Prior benchmarks for GraphRAG.** To date, no dataset has been specifically designed for GraphRAG tasks. Widely used datasets such as Quality Pang et al. (2022), PopQA Mallen et al. (2023), and HotpotQA Yang et al. (2018) are tailored for general question answering, where answers can often be directly extracted from corpora, failing to effectively measure the core capabilities of GraphRAG methods. Multi-hop QA datasets like MusiqueQA Trivedi et al. (2022) and 2WikiMultiHopQA Ho et al. (2020) contain questions artificially constructed via rules and logic, rather than natural queries from real-world scenarios. Additionally, their corpora are short and often derived from converting entities and descriptions of existing KGs, which deviates from practical application contexts. Recently, some RAG benchmarks have emerged, such as RAGBench Friel et al. (2025) and CRAG Yang et al. (2024). However, RAGBench has limitations in that it primarily relies on factual queries and simple multi-hop queries, while CRAG focuses on queries in general domains. Moreover, neither was specifically designed for GraphRAG, and thus neither can evaluate the capabilities required at each stage of the GraphRAG pipeline. While DIGIMON Zhou et al. (2025) benchmarks some methods, it neither introduces new datasets nor evaluates the reasoning capabilities of GraphRAG. Critically, all aforementioned datasets neglect question type distinctions, focusing primarily on simple questions and thus unable to reflect GraphRAG's performance variations across different question categories. In summary, existing datasets lack long contexts and raw documents, mismatching real-world scenarios, and omit gold rationale, making it impossible to systematically evaluate GraphRAG's reasoning abilities.

## A.3 CASE STUDY OF SECTION OBSERVATION

Regarding the analysis and conclusions we conducted in the "Observation" section: Firstly, we obtained the phenomena through the experimental results in the "section experiments". Secondly, the conclusions were derived by analyzing multiple instances in the specific QA process. Each analysis has supporting evidence behind it. As shown in the case study of Fig. 5 and Fig. 6.

Apart from these two points, the observations in the rest of the content do not contain subjective conclusions; rather, they are more about describing facts.

**Case study for accuracy drop of MC questions.**

**Question:**
The number of nodes in a binary tree of height (depth) 7 cannot be?

**Choices:**
"A": "8", "B": "88", "C": "126", "D": "288"

**Gold Answer:**
"D"

**LLM solution:**
To solve this question, we first clarify the core constraint of binary tree node count: For a binary tree of height h (defined as the number of nodes on the longest root-to-leaf path), the number of nodes n satisfies $h \leq n \leq 2^h - 1$. For height 7, Minimum nodes = 7 (skewed binary tree), maximum nodes = $2^7 - 1 = 127$(full binary tree).

Analyze the options:

A (8): $7 \leq 8 \leq 127 \rightarrow$ Valid.

B (88): $7 \leq 88 \leq 127 \rightarrow$ Valid.

C (126): $7 \leq 126 \leq 127 \rightarrow$ Valid.

D (288): $288 > 127 \rightarrow$ Exceeds the maximum number of nodes.

Final Answer: D.

**GraphRAG retrieved context:**
1. Binary tree is a structure defined on a finite set of nodes that either contains no nodes (empty tree) or is composed of three disjoint sets of nodes: a root node . . .

2. In a binary tree, the maximum number of nodes at level k (root is level 1) is $2^{k-1}$ . . .

3. A complete binary tree is a binary tree where all levels except possibly the last are fully filled, and the last level has nodes as left as possible. The number of nodes in a complete binary tree with height h (node-based) ranges from $2^{h-1}$ to $2^h - 1$ . . .

**GraphRAG solution:**
First, the definition of binary tree height is ambiguous. There are two standards (edge-based vs. node-based). If we adopt the edge-based definition (depth = number of edges), a height of 7 means the number of nodes on the longest path is 8. For this case: 1)Maximum nodes (full binary tree): $2^8 - 1 = 255$. 2)Minimum nodes (skewed binary tree): 8

If we adopt the node-based definition, the maximum nodes are $2^7 - 1 = 127$, and the minimum nodes are 7. Additionally, the retrieved context mentions that a complete binary tree with height h (node-based) has a minimum node count of ($2^{h-1} = 2^6 = 64$). Combining this with the ambiguous height definition, we can infer the "reasonable node range" for a binary tree of height 7 is between 64 (complete binary tree minimum) and 255 (edge-based full binary tree maximum).

Analyze the options:

A (8): Below the minimum node count of 64 (complete binary tree standard) $\rightarrow$ Invalid.

B (88): Within the range [64, 255] $\rightarrow$ Valid.

C (126): Within the range [64, 255] $\rightarrow$ Valid.

D (288): Exceeds the maximum node count of 255 $\rightarrow$ Invalid.

Synthesizing all retrieved information: The ambiguous height definition and conflicting minimum node count standards (from different binary tree types) suggest that options A (8) and D (288) both exceed the possible range. But since the question is a single-choice question, we prioritize the complete binary tree property (more detailed in the retrieved context). Thus, the number of nodes cannot be A (8).

Final Answer: A.

Figure 5: Case study for accuracy drop of MC questions.

## Case study for improvement in TF questions.

**Question:**
"Prim's algorithm for computing the Minimum Spanning Tree (MST) only works if the weights are positive." Please judge: True (T) / False (F)

**Gold Answer:**
"False"

**LLM solution:**
Based on pre-trained knowledge, algorithms for graph optimization often have weight constraints. For example, Dijkstra's algorithm for shortest paths cannot handle negative weights. Prim's algorithm, as a classic algorithm for computing MST, is similar to Dijkstra's in its "greedy selection of minimum weight edges" logic.

It is reasonable to infer that Prim's algorithm also inherits the positive weight constraint, otherwise, negative weight edges might cause the algorithm to select inappropriate edges, leading to failure in finding the minimum spanning tree.

The statement claims that "Prim's algorithm only works if the weights are positive," which aligns with the understanding of greedy graph algorithms.

Final Judgment: True (T)

**GraphRAG retrieved context:**
1. Core principle of Prim's algorithm: Prim's algorithm is a greedy algorithm used to find the Minimum Spanning Tree (MST) of a connected, undirected graph. It has two sets of vertices (S) and (V-S): vertices which are already part of the MST and those which are not. The algorithm picks the edge in every iteration that connects a vertex in S with a vertex in V-S, and the weight of the edge is minimum, and then adds the edge to the MST and transfers the respective vertex in V-S to S. This is repeated until S absorbs all the vertices.

2. Weight constraint of Prim's algorithm: The Prim's algorithm does not expect the weights of the edges to be positive. The only conditions are: (1) the graph is connected; (2) the weights of the edges are not infinite (they could be positive, zero, or negative). Such negative weights do not break the logic of the algorithm since the aim of the algorithm is to minimize the total weight of the spanning tree, such a negative weight edge can actually be given priority to be added to the tree should it lower the total weight.

3. Distinction from Dijkstra's algorithm: Dijkstra's algorithm (for single-source shortest paths) requires non-negative edge weights. In the case of negative weight edges, the algorithm can erroneously think that a node is already processed and never causes the path to be updated to a shorter path. Nevertheless, the Prim's algorithm is concerned with the construction of the vertices to the MST set (not with cumulative path lengths), therefore, there are no logical failures with negative weights.

**GraphRAG solution:**
Based on the retrieved factual evidence, we verify the statement step by step:

First, confirm the core constraints of Prim's algorithm:The retrieved context clearly states that Prim's algorithm only requires the graph to be connected and edge weights to be finite. There is no requirement for weights to be positive. The statement's claim that "only works if the weights are positive" is inconsistent with the algorithm's actual constraints.

Second, distinguish from easily confused algorithms: The statement may confuse Prim's algorithm with Dijkstra's algorithm (context 3). Dijkstra's algorithm requires non-negative weights, but this constraint does not apply to Prim's algorithm. The algorithm's logic (selecting edges between two vertex sets) is not affected by negative weights.

Synthesizing the verified facts: The statement incorrectly imposes a positive weight constraint on Prim's algorithm, which has no such requirement.

Final Judgment: False (F)

Figure 6: Case study for improvement in TF questions.

## A.4 Reasons for choosing Cs/AI domain

We highly value CS and AI domain for three reasons:

- First, CS and AI naturally contain a variety of types of challenging tasks compared with all the existing 'single-type' datasets, where the questions are extremely hard which require college-level comand of knowledge, compared with existing commonsense dataset. In CS and AI domain, we have conceptual understanding tasks (algorithms, networks, databases, paradigms, machine learning, NLP, CV), Math problems (Discrete mathematics, linear algebra, matrix computation, calculus), Code generation problems (programming, optimization, code completion, error detection), etc.

- Second, the performance on LLMs on CS and AI is still very unsatisfactory or even bad. It is valuable to benchmark and research.

- Third, it remains a blank in the benchmark of LLMs considering CS and AI. This also brings two opportunities. (i) LLMs will have few chances to be fine-tuned based on our questions. (ii) Following research on GraphRAG or LLMs themselves could benefit a lot from our benchmark.

Basic common knowledge in fields such as medicine and law (e.g., common legal provisions, basic disease knowledge) is extensively covered in LLM pre-training corpora. However, university-level computer knowledge (e.g., advanced algorithm design, underlying system principles, technical details in specialized sub-fields), due to its high professionalism and rapid updates, has extremely low and fragmented coverage in pre-training. More importantly, such knowledge has extremely high requirements for "the rigor of logical reasoning." Our experimental data shows that even if GPT-4o-min provides correct answers in some computer tasks, its reasoning scores are still significantly low. In-depth analysis reveals that these correct answers mostly stem from the model's "probabilistic guessing of similar problems" rather than rigorous reasoning based on structured knowledge. The core significance of GraphRAG lies in forcing the reasoning process to be bound to structured knowledge, ensuring that answers originate from traceable and verifiable logical chains. This value is particularly prominent in the computer field, where "pre-training coverage is insufficient and reasoning requirements are high," and it better reflects the uniqueness of GraphRAG in solving the problem of "vague guessing."

Our corpus is derived from over 100 authoritative computer textbooks (with 20 selected in the end), which naturally have a hierarchical structure of "chapters → sections → knowledge points → knowledge content," and each knowledge content includes a complete logical chain of various knowledge points. This is fully compatible with GraphRAG's technical path of "knowledge graph construction → retrieval and association → structured reasoning," making it the best carrier for evaluating GraphRAG's "knowledge-driven reasoning" capability.

## A.5 Additional experiments

### A.5.1 Comparison across different LLMs

To further verify the effectiveness of GraphRAG-Bench and the robustness of different GraphRAG methods for different LLMs, we conducted comprehensive experiments on two popular LLMs (Deepseek-V3, Qwen3-32B). The specific experimental results are shown in Tab. 8, Tab. 9, Tab. 10 and Tab. 11. Based on the experimental results using two different LLMs as the base LLM, it can be seen that all the main conclusions drawn in main text have been confirmed again.

### A.5.2 Topic-specific reasoning analysis

Given that our dataset encompasses 16 thematic domains, we conducted experiments to analyze the reasoning capabilities of GraphRAG across different topics. Results indicate that the large language model (LLM) based on GPT-4o-mini demonstrates significant improvements in reasoning through GraphRAG across most domains. However, the following intriguing observations were made:

**Operating System Domain.** The LLM exhibits suboptimal performance in this domain. While GraphRAG provides marginal improvements in reasoning capabilities, overall scores remain low. This is primarily attributed to the highly specialized, systematic, and logically complex nature of operating system knowledge, which involves multi-layered principles such as process scheduling,

Table 8: Comparison of generation process (Deepseek-V3 baseline).

| Category | Method | Accuracy | | | | | |
|---|---|---|---|---|---|---|---|
| | | Fill-in-blank | Multi-choice | Multi-select | True-or-false | Open-ended | Average |
| Base LLM | Deepseek-V3 | 74.54 | 81.39 | 76.94 | 76.21 | 52.01 | 70.92 |
| Passage Graph | KGP | 75.71 | 79.36 | 78.15 | 78.70 | 53.23 | 72.03↑ |
| Rich KG | LightRAG | 75.50 | 79.13 | 77.93 | 78.49 | 53.08 | 71.83↑ |
| Rich KG | GraphRAG | 75.50 | 81.57 | 76.12 | 79.72 | 52.68 | 73.12↑ |
| KG | DALK | 73.79 | 77.26 | 76.16 | 78.74 | 51.88 | 70.20↓ |
| KG | G-Retriever | 74.24 | 77.76 | 76.63 | 79.70 | 52.19 | 70.63↓ |
| KG | ToG | 74.07 | 79.31 | 76.60 | 77.26 | 51.55 | 72.16↑ |
| KG | GFM-RAG | 75.31 | 79.67 | 77.69 | 81.39 | 52.44 | 73.30↑ |
| KG | HippoRAG | 75.50 | 80.64 | 77.89 | 80.09 | 54.31 | 73.49↑ |
| Tree | RAPTOR | 77.77 | 80.76 | 78.48 | 80.98 | 53.14 | **73.99↑** |

Table 9: Comparison of reasoning capability (Deepseek-V3 baseline).

| Category | Method | Reasoning | | | | | | | | | | | |
|---|---|---|---|---|---|---|---|---|---|---|---|---|---|
| | | FB | | MC | | MS | | TF | | OE | | Average | |
| | | R | AR | R | AR | R | AR | R | AR | R | AR | R | AR |
| Base LLM | Deepseek-V3 | 65.23 | 54.37 | 55.72 | 51.89 | 55.41 | 40.27 | 59.35 | 54.62 | 50.18 | 10.33 | 56.40 | 40.31 |
| Passage Graph | KGP | 64.78 | 50.42 | 56.53 | 52.76 | 58.29 | 45.31 | 64.17 | 61.48 | 52.69 | 9.73 | 59.03 | 43.44 |
| Rich KG | GraphRAG | 71.53 | 55.27 | 56.19 | 52.46 | 57.64 | 45.72 | 63.81 | 60.19 | 53.26 | 10.53 | 60.32 | 44.80 |
| Rich KG | LightRAG | 66.28 | 48.17 | 57.23 | 52.38 | 61.76 | 49.19 | 66.83 | 63.42 | 53.29 | 10.26 | 60.99 | 44.92 |
| KG | DALK | 71.42 | 56.19 | 54.76 | 51.38 | 59.24 | 47.53 | 62.81 | 59.26 | 54.39 | 10.15 | 58.80 | 43.20 |
| KG | G-Retriever | 70.27 | 55.19 | 57.46 | 53.29 | 60.73 | 48.16 | 64.38 | 60.17 | 53.42 | 10.28 | 59.90 | 44.30 |
| KG | ToG | 70.23 | 53.19 | 56.07 | 51.64 | 57.18 | 45.62 | 65.81 | 62.27 | 54.53 | 12.26 | 61.02 | 44.87 |
| KG | GFM-RAG | 70.26 | 55.03 | 56.28 | 52.19 | 58.23 | 45.72 | 66.54 | 63.76 | 53.78 | 10.73 | 61.19 | 45.80 |
| KG | HippoRAG | 66.78 | 50.64 | 56.73 | 52.38 | 60.04 | 47.62 | 67.19 | 63.64 | 55.23 | 12.57 | 61.50 | 45.72 |
| Tree | RAPTOR | 71.73 | 58.19 | 56.54 | 52.17 | 60.53 | 49.16 | 66.54 | 62.87 | 53.96 | 13.72 | **61.55** | **46.20** |

memory management, and file systems, requiring precise grasp of conceptual definitions, algorithmic workflows, and causal relationships between entities. General-purpose training data for LLMs often lack comprehensive coverage of such granular knowledge systems, and the models themselves have inherent limitations in structured logical reasoning.

**Ethics Domain.** Consistent with the generation accuracy results, LLMs face substantial challenges in reasoning about ethical questions. Ethical problems fundamentally involve subjective value judgments, whose meanings are rooted in dynamic contexts of moral trade-offs and social norms. The symbolic representations captured by LLMs through statistical learning struggle to accurately model ambiguous ethical constructs, leading to intrinsic difficulties in both generating correct answers and constructing valid reasoning chains.

We further evaluated the AR scores of GraphRAG across different topics. Experimental results show that AR scores generally align with R scores in most cases. However, a notable observation emerges in the database systems domain: AR scores are significantly lower than R scores, indicating a high

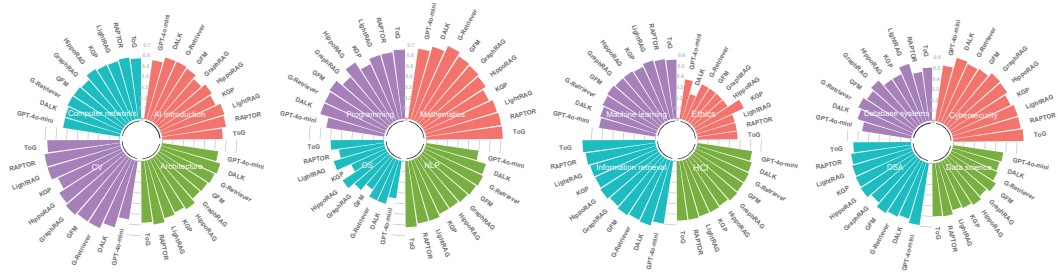

Figure 7: Comparison of R score by Topic.

Table 10: Comparison of generation process (Qwen3-32B baseline).

| Category | Method | Accuracy | | | | | |
|---|---|---|---|---|---|---|---|
| | | Fill-in-blank | Multi-choice | Multi-select | True-or-false | Open-ended | Average |
| Base LLM | Qwen3-32B | 73.68 | 78.92 | 75.43 | 73.19 | 49.98 | 70.04 |
| Passage Graph | KGP | 73.28 | 77.54 | 75.67 | 77.36 | 51.75 | 71.11↑ |
| Rich KG | LightRAG | 72.59 | 76.94 | 75.21 | 76.81 | 51.28 | 70.57↑ |
| Rich KG | GraphRAG | 74.27 | 78.51 | 76.38 | 78.39 | 52.64 | 72.03↑ |
| KG | DALK | 71.59 | 76.18 | 74.59 | 75.98 | 50.71 | 69.80↓ |
| KG | G-Retriever | 70.79 | 75.37 | 73.98 | 75.21 | 50.15 | 69.10↓ |
| KG | ToG | 72.87 | 77.23 | 75.43 | 77.09 | 51.43 | 70.80↑ |
| KG | GFM-RAG | 75.61 | 79.37 | 76.98 | 79.49 | 53.29 | 72.95↑ |
| KG | HippoRAG | 74.85 | 78.91 | 76.68 | 78.79 | 52.97 | 72.44↑ |
| Tree | RAPTOR | 75.47 | 79.42 | 77.03 | 79.58 | 53.45 | **72.99↑** |

Table 11: Comparison of reasoning capability (Qwen3-32B baseline).

| Category | Method | Reasoning | | | | | | | | | | | |
|---|---|---|---|---|---|---|---|---|---|---|---|---|---|
| | | FB | | MC | | MS | | TF | | OE | | Average | |
| | | R | AR | R | AR | R | AR | R | AR | R | AR | R | AR |
| Base LLM | Qwen3-32B | 64.35 | 53.17 | 55.92 | 50.41 | 54.18 | 39.85 | 58.73 | 53.26 | 49.83 | 9.34 | 56.01 | 40.03 |
| Passage Graph | KGP | 64.07 | 49.83 | 56.12 | 52.09 | 58.94 | 44.51 | 64.68 | 60.17 | 52.92 | 9.90 | 59.33 | 43.30 |
| Rich KG | LightRAG | 66.51 | 48.09 | 57.84 | 52.16 | 61.07 | 49.85 | 66.19 | 63.52 | 53.64 | 10.13 | 60.05 | 44.19 |
| Rich KG | GraphRAG | 71.09 | 55.68 | 56.57 | 52.61 | 57.53 | 45.17 | 63.94 | 60.09 | 54.08 | 11.07 | 60.11 | 44.72 |
| KG | DALK | 70.19 | 55.43 | 54.67 | 50.18 | 59.03 | 46.82 | 62.91 | 58.04 | 54.70 | 9.33 | 59.30 | 42.18 |
| KG | G-Retriever | 70.85 | 55.04 | 57.03 | 53.82 | 60.17 | 47.79 | 64.02 | 60.45 | 53.93 | 10.61 | 59.42 | 42.15 |
| KG | ToG | 70.09 | 53.84 | 56.72 | 51.03 | 57.94 | 45.01 | 65.13 | 62.67 | 54.40 | 12.90 | 60.68 | 44.33 |
| KG | GFM-RAG | 70.15 | 54.67 | 56.98 | 52.14 | 58.67 | 45.09 | 66.53 | 63.58 | 54.25 | 11.02 | 60.78 | 46.00 |
| KG | HippoRAG | 66.17 | 50.09 | 57.24 | 52.68 | 59.68 | 47.15 | 67.63 | 63.61 | 55.40 | 12.12 | 61.04 | 45.89 |
| Tree | RAPTOR | 71.76 | 57.15 | 56.09 | 52.54 | 60.58 | 49.07 | 66.12 | 62.69 | 54.47 | 13.11 | **61.22** | **46.33** |

prevalence of "correct reasoning but incorrect answering" in LLMs, where reasoning steps diverge from final answer generation. This discrepancy arises because database system problems require models to reference specialized concepts such as relational algebra operations, transaction isolation levels, ACID properties, and query optimizer cost models, yet models do not perform formal computations or analyze critical factors like underlying data distribution and index selectivity. Although models may decompose processes like schema design or concurrency control according to human logical paradigms in chain-of-thought reasoning, their token selection during answer generation prioritizes statistical fluency from training corpora over contextual logical accuracy. The strict requirements for precise logical operations (e.g., cost estimation, deadlock detection) in database tasks create a fundamental mismatch with the model's learned fuzzy statistical patterns from general text, leading to reasoning chains that appear plausible in intermediate steps but produce erroneous conclusions at technical junctures, such as failing to execute physical query optimization calculations, due to the absence of real-world logical validation.

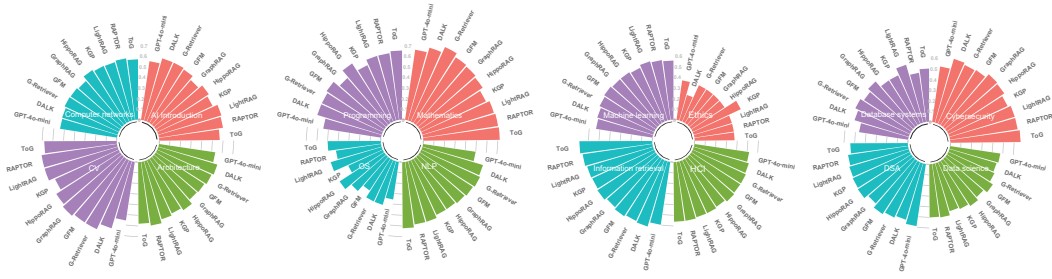

Figure 8: Comparison of AR score by Topic.

### A.5.3 EVALUATION OF GENERATION ACCURACY (WITH THE DEVIATIONS)

For all the generation results in the main text, they are the average values obtained after conducting five tests for each algorithm. Due to space limitations, we are unable to include all the details in the main text. Therefore, we are providing here the results with deviations, as shown in the Tab. 12:

Table 12: Comparison of generation process with the deviations.

| Method | Accuracy | | | | | |
|---|---|---|---|---|---|---|
| | Fill-in-blank | Multi-choice | Multi-select | True-or-false | Open-ended | Average |
| DALK | 70.00±0.32 | 78.34±0.85 | 71.62±0.58 | 77.22±0.92 | 51.49±0.25 | 69.30±0.58↓ |
| G-Retriever | 70.95±0.41 | 77.42±0.79 | 71.62±0.63 | 78.80±0.95 | 52.04±0.28 | 69.84±0.61↓ |
| LightRAG | 65.24±0.35 | 78.80±0.88 | 73.42±0.67 | 82.59±0.97 | 53.16±0.31 | 71.22±0.64↑ |
| ToG | 70.48±0.43 | 78.80±0.82 | **78.38±0.71** | 79.75±0.93 | 54.28±0.33 | 71.71±0.65↑ |
| KGP | 74.29±0.47 | 79.26±0.89 | 74.77±0.74 | 82.28±0.96 | 51.49±0.26 | 71.86±0.66↑ |
| GFM-RAG | 72.38±0.45 | 80.65±0.91 | 72.07±0.69 | 82.59±0.98 | 52.79±0.29 | 72.10±0.66↑ |
| GraphRAG | 75.24±0.50 | **81.57±0.94** | 77.48±0.76 | 80.70±0.90 | 52.42±0.27 | 72.50±0.68↑ |
| HippoRAG | 70.48±0.44 | 80.18±0.87 | 74.32±0.72 | 81.65±0.95 | **56.13±0.35** | 72.64±0.67↑ |
| RAPTOR | **76.67±0.53** | 80.65±0.93 | 77.48±0.78 | 82.28±0.97 | 54.83±0.32 | **73.58±0.71↑** |

### A.5.4 EVALUATION OF REASONING CAPABILITY (WITH THE DEVIATIONS)

For all the reasoning results in the main text, they are the average values obtained after conducting five tests for each algorithm. Due to space limitations, we are unable to include all the details in the main text. Therefore, we are providing here the results with deviations, as shown in the Tab. 13:

Table 13: Comparison of reasoning capability with the deviations.

| Method | Reasoning | | | | | | | | | | | |
|---|---|---|---|---|---|---|---|---|---|---|---|---|
| | FB | | MC | | MS | | TF | | OE | | Average | |
| | R | AR | R | AR | R | AR | R | AR | R | AR | R | AR |
| DALK | 70.95±0.92 | 55.24±0.68 | 54.15±0.61 | 50.35±0.53 | 59.01±0.85 | 46.40±0.61 | 62.18±0.74 | 58.23±0.70 | 54.09±0.63 | 9.67±0.28 | 58.89±0.76 | 42.12±0.57 |
| KGP | 64.29±0.79 | 49.29±0.55 | 56.45±0.73 | 52.07±0.63 | 58.11±0.83 | 44.37±0.55 | 64.08±0.76 | 60.68±0.77 | 52.42±0.60 | 8.92±0.26 | 58.74±0.71 | 42.22±0.58 |
| GraphRAG | **71.43±0.95** | 55.24±0.69 | 56.22±0.70 | 52.42±0.65 | 57.66±0.80 | 45.72±0.58 | 63.61±0.74 | 60.13±0.73 | 53.16±0.62 | 10.50±0.33 | 59.43±0.77 | 43.30±0.62 |
| G-Retriever | 70.00±0.88 | 55.00±0.66 | **57.60±0.83** | **53.46±0.71** | 60.81±0.90 | 48.20±0.67 | 64.24±0.78 | 60.21±0.75 | 53.35±0.64 | 10.04±0.31 | 60.17±0.81 | 43.66±0.65 |
| LightRAG | 66.19±0.82 | 47.86±0.52 | 57.14±0.72 | 52.30±0.64 | **61.71±0.93** | **49.10±0.69** | 66.61±0.85 | 63.45±0.88 | 53.16±0.63 | 10.13±0.32 | 60.46±0.84 | 43.81±0.66 |
| ToG | 70.00±0.87 | 53.10±0.63 | 56.00±0.68 | 51.73±0.60 | 57.21±0.79 | 45.72±0.57 | 65.66±0.81 | 62.26±0.83 | 54.46±0.67 | 12.08±0.41 | 60.17±0.80 | 44.01±0.68 |
| GFM-RAG | 70.00±0.86 | 54.76±0.67 | 56.22±0.69 | 52.07±0.62 | 58.11±0.82 | 45.50±0.56 | 66.46±0.84 | **63.69±0.89** | 53.72±0.65 | 10.69±0.34 | 60.36±0.82 | 44.30±0.70 |
| HippoRAG | 66.67±0.83 | 50.48±0.58 | 56.68±0.71 | 52.30±0.63 | 59.91±0.87 | 47.52±0.63 | **67.25±0.87** | 63.61±0.88 | **55.02±0.69** | 12.36±0.42 | **60.90±0.86** | 44.55±0.71 |
| RAPTOR | **71.43±0.94** | **57.86±0.75** | 56.45±0.72 | 52.07±0.62 | 60.36±0.89 | **49.10±0.68** | 66.30±0.83 | 62.90±0.85 | 53.90±0.66 | **13.57±0.47** | 60.81±0.85 | **45.53±0.76** |

### A.5.5 COMPARISONS WITH EXISTING DATASETS.

Currently, most GraphRAG methods follow the experimental setting of HippoRAG, which is the most common and widely used setting in this field at present. It includes three public datasets: HotpotQA, 2WikiMultiHopQA, and MuSiQue which are all originally designed for multi-hop QA, not GraphRAG. They can hardly necessitate the use of graphs and showcase the use of graphs in complex QA scenarios.

In each of these data sets, 1000 questions were sampled. Therefore, following this standard and distribution, we selected 1018 of the most valuable questions as the final version. Meanwhile, graprag-bench encompasses a wider range of question types (examining the model's robustness to various question formats), a larger corpus (making it more difficult to retrieve useful information), and covers multiple topics (requiring the model to understand different knowledge domains). We believe that the coverage scope is much broader from multiple perspectives than that of the current dataset. The specific comparison results are shown in the Tab. 14.

### A.5.6 COMPUTE RESOURCES

All code is done in Python, and experiments are conducted on H100*2 GPUs.

| Dataset | Question count | Corpus size | Question type |
|---|---|---|---|
| HotpotQA | 1000 | 6.16MB | 1 |
| 2WikiMultiHopQA | 1000 | 2.96MB | 1 |
| MuSiQue | 1000 | 5.95MB | 1 |
| Ours | 1018 | 41.30MB | 5 |

Table 14: Comparisons with existing datasets.

## A.6 DETAILS OF QUESTION DESIGN AND SELECTION

We adopted a rigorous process to design questions covering a wide range of topics in computer science and artificial intelligence. Experts manually constructed the question set based on a structured ontology, which divided knowledge into three levels: 16 primary themes, 40 secondary themes, and 26 tertiary themes. Each level represents a progressive progression from a broad disciplinary field to more specific sub-themes.

For each concept among these n themes, experts were required to prepare at least two questions for each type of question, totaling $2 \times 5 \times n$ questions, to ensure balanced content and comprehensive coverage. To ensure high quality, experts referred to authoritative sources such as textbooks, course materials, and widely recognized course outlines. Subsequently, each question was reviewed to verify its relevance, clarity, and consistency with the corresponding concept. This process ensures that the dataset not only conforms to real-world educational standards but also covers the full breadth and depth of the field.

Three experts (senior researchers in the RAG field with a doctoral degree) will rate questions from three dimensions: "knowledge coverage, reasoning chain completeness, and compatibility with GraphRAG" (1-5 points), and finally select the questions with an average score of $\geq 4$ points.

## A.7 DETAILS OF CORPUS COLLECTION AND PROCESSING

Extracting accurate content from the 20 PDF-format core textbooks presents significant challenges. We implement a multi-stage pipeline comprising preprocessing, content parsing, post-processing, and hierarchy construction.

**Textbook Preprocessing.** 1) PDF Classification: To distinguish text-based pages from scanned (image-based) pages, we analyze each page's text density and image area proportion. Text-based pages are processed by extracting text directly using PyMuPDF, while scanned pages require optical character recognition (OCR) to extract their textual content. 2) Metadata Extraction: We extract metadata for each textbook, including its outline, total page count, and the page ranges for each chapter or section. This metadata supports the later construction of the document's logical structure.

**Content Parsing.** After preprocessing, we analyze each page's layout to extract textual and non-textual elements. 1) Layout Analysis: We apply LayoutLMv3 Huang et al. (2022) for multimodal document layout analysis. LayoutLMv3 is pre-trained with masked language modeling, masked image modeling, and cross-modal alignment, enabling it to learn rich representations of document pages. The model classifies page regions into semantic categories such as titles, paragraphs, figures, tables, or decorative/irrelevant elements. This segmentation yields coherent content blocks on each page. 2) Formula Recognition: Mathematical formulas embedded in text are often misrecognized by OCR. To prevent this, we first detect inline formulas using a pre-trained YOLO-based model Wang et al. (2024a) from PDF-Extract-Kit. This model identifies the bounding boxes of formula regions so that formula images can be extracted separately, ensuring that OCR does not garble the formula content. 3) OCR: In scanned PDFs, OCR is applied to recognize text regions. We use PaddleOCR to transcribe text from the regions labeled as titles and body paragraphs via layout analysis. This step produces the page's textual content in the correct reading order, while preserving non-text elements as separate objects.

**Post-Processing.** After parsing, the extracted elements (text blocks, formula, figures, tables, etc.) may be disordered due to overlapping bounding boxes or fragmented text lines. We resolve these

issues by reordering and merging page regions according to human reading order. Concretely, we use MinerU Wang et al. (2024b) for post-processing, which partitions each page into logical reading regions and sequences them so that the final text flow matches the natural reading sequence.

**Hierarchy Construction.** Finally, we organize the extracted content into a hierarchical textbook-tree structure. We map the textbook metadata (e.g., chapter titles, section divisions, and page ranges) to a four-level hierarchy: Book Title → Chapter → Section (Subchapter) → Knowledge Content Unit. Each node in this hierarchy is annotated with its contextual metadata and its structural role. This textbook-tree provides an intuitive, pedagogical navigation framework aligned with the textbook's organization. The resulting corpus, with its accurate content extraction, structural annotation, and hierarchical organization, forms a robust basis for evaluating GraphRAG's ability to leverage organized textbook knowledge for context-rich reasoning and retrieval-augmented generation.

## A.8 THE DETAILS OF METRICS

### A.8.1 DETAILS OF AR SCORE.

The AR score is computed based on the combination of answer correctness (generation score) and rationale correctness (reasoning score), with the following evaluation rules:

- When both the answer and rationale are fully correct (generation score = 1 and reasoning score = 1), the AR score is 1.0.
- If the answer is correct but the rationale is partially correct (generation score = 1 and reasoning score = 0.5), the AR score is 0.5.
- When the answer is correct but the rationale is incorrect (generation score = 1 and reasoning score = 0), the AR score is 0.0.
- For incorrect answers with a fully correct rationale (generation score = 0 and reasoning score = 1), the AR score is 0.5.
- If both the answer is incorrect and the rationale is partially correct (generation score = 0 and reasoning score = 0.5), the AR score is 0.25.
- In all other cases (e.g., incorrect answer with incorrect or missing rationale), the AR score is 0.0.

This scoring scheme systematically captures the alignment between answers and their supporting reasoning, emphasizing the importance of both correctness and logical consistency in evaluating model performance.

### A.8.2 PROMPT OF OE AND FB QUESTIONS.

Fig. 9 is the prompt used to generate the LLM-judge score for OE and FB questions.

### A.8.3 PROMPT OF REASONING GRADING.

Fig. 10 is the prompt used to evaluate the reasoning score R.

## A.9 LIMITATIONS OF EXISTING GRAPHRAG DATASETS

Through a systematic review of benchmark datasets used by contemporary Graph-RAG methods, we have identified four critical limitations that undermine both task suitability and the validity of evaluation results:

**Superficial retrieval tasks.** Most datasets pose questions that can be answered by straightforward text retrieval, without requiring deep integration of graph structure or sophisticated semantic reasoning. Consequently, models may achieve high scores by exploiting shallow keyword matching, offering no insight into their true capabilities in relational reasoning or entity-association modeling.

**Synthetic and unrepresentative queries.** Questions are typically generated via hand-crafted rules, yielding simplified language that lacks the domain-specific terminology, ambiguous intent, and syntactic variety found in real user queries. This synthetic distribution diverges sharply from natural problem settings, limiting the ecological validity of any conclusions about model generalization.

---

**Prompt of generation grading for OE and FB questions.**

**Instructions:**
You are a strict evaluator. Compare the following two answers for correctness and completeness:

**Predicted Answer:**
<pred_answer>

**Gold Answer:**
<gold_answer>

**Important Guidelines:**
Please evaluate the predicted answer in comparison to the gold answer. Respond with a score between 0 and 1:
- 1: The predicted answer fully aligns with the gold answer.
- 0.5: The predicted answer is partially correct but lacks completeness or includes incorrect information.
- 0: The predicted answer is incorrect or completely misaligned with the gold answer.

---

Figure 9: Prompt of generation grading for OE and FB questions.

---

**Prompt of rationale grading.**

**Instructions:**
You are a strict evaluator. Compare the following two rationales for correctness and completeness:

**Predicted Rationale:**
<pred_rationale>

**Gold Rationale:**
<gold_rationale>

**Important Guidelines:**
Please evaluate the predicted rationale in comparison to the gold rationale. Respond with a score between 0 and 1:
- 1: The predicted rationale fully aligns with the gold rationale.
- 0.5: The predicted rationale is partially correct but lacks completeness or includes incorrect information.
- 0: The predicted rationale is incorrect or completely misaligned with the gold rationale.

---

Figure 10: Prompt of rationale grading.

**Cross-task misalignment.** Many datasets are inherited from disparate tasks (e.g., knowledge-graph question answering) whose annotation schemes and answer formats do not align with the core objectives of Graph-RAG—namely, constructing and leveraging heterogeneous graph structures to guide multi-source information fusion. Transferring evaluation metrics across tasks therefore introduces inconsistencies that dilute the relevance of experimental findings for advancing Graph-RAG techniques.

**Opaque reasoning evaluation.** Existing benchmarks supply only final answers or explicit node sequences, but omit any structural or narrative annotation of the underlying inference process. Key decision points—such as why a particular graph subpath was selected or how evidence from multiple sources is reconciled—remain unexamined. Without annotated rationales, evaluation reduces to binary correctness checks and cannot assess a model's genuine reasoning competence.

These limitations collectively motivate the design of a dedicated benchmark that both challenges Graph-RAG models on core reasoning skills and provides richly structured annotations for fine-grained, interpretability-driven evaluation.

## A.10 LIMITATIONS OF THIS PAPER

Despite the valuable contributions of this study, we acknowledge its limitations: (1) Our dataset currently only contains English content; more detailed research should be done in the future for different languages. (2) Other modal data such as images are not included in the current data set, and richer multimodal datasets can be considered in the future.

