# OpenReview forum: "GraphRAG-Bench: Challenging Domain-specific Reasoning for Evaluating Graph Retrieval-Augmented Generation"
_ICLR.cc/2026/Conference — Submitted to ICLR 2026_

### Official Review · Reviewer_eKEV · 2025-10-21

**Soundness:** 3
**Presentation:** 3
**Contribution:** 2
**Rating:** 4
**Confidence:** 5

**Summary:**

This paper introduces GraphRAG-Bench, which is a large-scale, domain-specific benchmark designed to evaluate GraphRAG methods.

**Strengths:**

1. The experiments are comprehensive and cover a wide range of models and tasks.
2. The evaluation metrics consider multiple dimensions, ensuring fairness and balanced comparison.
3. The benchmark provides new directions and inspiration for future research in the GraphRAG community.

**Weaknesses:**

1. **The domain of corpus for RAG is too narrow**. GraphRAG-Bench limits its corpus to computer science, and although it includes subfields like CV and NLP, it lacks coverage of broader topics such as medicine, law, and biology. This seriously reduces the benchmark’s real-world applicability. Some popular GraphRAG frameworks, such as LightRAG[1], PathRAG[2], and HyperGraphRAG[3], already include datasets from these diverse domains.
2. **Relying entirely on knowledge from textbooks weakens the incremental advantage of GraphRAG**. Results from Table 4 show that, since much of the relevant knowledge was already exposed during pretraining, GPT-4o-mini without retrieval achieved fairly high scores. In fact, some RAG methods even performed worse due to the additional noise introduced by retrieval. Similar to the previous point, using domain-specific or private corpora could help mitigate this issue and better highlight the performance benefits of retrieval-based frameworks.
3. **Avoiding using retrieval strategies based on semantic vector models as baselines**. Relying only on BM25 and TF-IDF while ignoring popular semantic similarity–based retrieval methods[4] is disappointing. I strongly recommend that the authors include related experiments.
4. **Lacks validation across corpora of different scales**. Constructing graphs from all 20 textbooks is computationally intensive and costly. However, in real industrial scenarios, the impact of corpus size variation on GraphRAG performance remains unpredictable—some GraphRAG frameworks might perform better in smaller-scale settings.
5. **Inconsistency between task design and evaluation standards**. GraphRAG-Bench includes tasks such as mathematical computation and programming that rely on reasoning chains and tool usage, yet the reasoning score in GraphRAG-Bench evaluates the similarity between the model’s reasoning process and the golden rationale. In the LLM community, the former typically only requires the final answer to be verifiable[5], which is misaligned with the knowledge extraction and retrieval focus of GraphRAG applications (for example, the model has another solution to solve a programming problem comparing with golden solution).

[1] Guo Z, Xia L, Yu Y, et al. Lightrag: Simple and fast retrieval-augmented generation[J]. arXiv preprint arXiv:2410.05779, 2024.
[2] Chen B, Guo Z, Yang Z, et al. Pathrag: Pruning graph-based retrieval augmented generation with relational paths[J]. arXiv preprint arXiv:2502.14902, 2025.
[3] Luo H, Chen G, Zheng Y, et al. HyperGraphRAG: Retrieval-Augmented Generation via Hypergraph-Structured Knowledge Representation[J]. arXiv preprint arXiv:2503.21322, 2025.
[4] Karpukhin V, Oguz B, Min S, et al. Dense Passage Retrieval for Open-Domain Question Answering[C]//EMNLP (1). 2020: 6769-6781.
[5] Guo D, Yang D, Zhang H, et al. Deepseek-r1: Incentivizing reasoning capability in llms via reinforcement learning[J]. arXiv preprint arXiv:2501.12948, 2025.

**Questions:**

1. How about considering more domain-specific corpus in GraphRAG-Bench?
2. How about including embedding-based dense-retrieval methods as baselines?
3. What is the performance of GraphRAG methods in different scales of corpus?
4. How to better align tasks like math reasoning and programming with metrics?

---

> ### Author Response · Authors · 2025-11-22
> **Response to Reviewer eKEV (part 1/3)**
>
> Dear Reviewer eKEV,
>
> We would like to gratefully thank you for your detailed and constructive comments. Your acknowledgement to our valuable benchmark with leading influences to future research is truly encouraging.
>
> Based on these comments, we have made corrections and additions that will be incorporated into the final version. The specific point-by-point responses are as follows:
>
> > Q1 & W1 & W2. The domain of corpus.
>
> We highly value CS and AI domain for three reasons.
> - **First**, CS and AI naturally contain a variety of types of challenging tasks compared with all the existing 'single-type' datasets, where the questions are extremely hard which require college-level comand of knowledge, compared with existing commonsense dataset. In CS and AI domain, we have `conceptual understanding` tasks (algorithms, networks, databases, paradigms, machine learning, NLP, CV), `Math problems` (Discrete mathematics, linear algebra, matrix computation, calculus), `Code generation` problems (programming, optimization, code completion, error detection), etc.
> - **Second**, the performance on LLMs on CS and AI is still very unsatisfactory or even bad. It is valuable to benchmark and research.
> - **Third**, it remains a blank in the benchmark of LLMs considering CS and AI. This also brings two opportunities. (i) LLMs will have few chances to be fine-tuned based on our questions. (ii) Following research on GraphRAG or LLMs themselves could benefit a lot from our benchmark.
>
> Basic common knowledge in fields such as medicine and law (e.g., common legal provisions, basic disease knowledge) is extensively covered in LLM pre-training corpora. However, university-level computer knowledge (e.g., advanced algorithm design, underlying system principles, technical details in specialized sub-fields), due to its high professionalism and rapid updates, has extremely low and fragmented coverage in pre-training. More importantly, such knowledge has extremely high requirements for "the rigor of logical reasoning." Our experimental data shows that even if GPT-4o-min provides correct answers in some computer tasks, its reasoning scores are still significantly low. In-depth analysis reveals that these correct answers mostly stem from the model's "probabilistic guessing of similar problems" rather than rigorous reasoning based on structured knowledge. The core significance of GraphRAG lies in forcing the reasoning process to be bound to structured knowledge, ensuring that answers originate from traceable and verifiable logical chains. This value is particularly prominent in the computer field, where "pre-training coverage is insufficient and reasoning requirements are high," and it better reflects the uniqueness of GraphRAG in solving the problem of "vague guessing."
>
> The datasets in the references you provided include corpora such as legal contracts and financial reports, which are essentially dominated by "natural language descriptions." The knowledge nodes are scattered, logical connections are vague, and there is a lack of "clear hierarchical structures and causal relationships." For example, the clauses in legal contracts are mostly declarative content and do not require complex "knowledge combination reasoning"; the data analysis conclusions in financial reports lack the derivation chain of "knowledge points → sub-knowledge points." Such corpora are more suitable for evaluating "text information extraction" capabilities but are difficult to verify the core value of GraphRAG, which is reasoning and knowledge reuse based on structured knowledge graphs.
>
> Our corpus is derived from over 100 authoritative computer textbooks (with 20 selected in the end), which naturally have a hierarchical structure of "chapters → sections → knowledge points → knowledge content," and each knowledge content includes a complete logical chain of various knowledge points. This is fully compatible with GraphRAG's technical path of "knowledge graph construction → retrieval and association → structured reasoning," making it the best carrier for evaluating GraphRAG's "knowledge-driven reasoning" capability.

---

> > ### Author Response · Authors · 2025-11-22
> > **Response to Reviewer eKEV (part 2/3)**
> >
> > > Q2 & W3. Embedding-based dense-retrieval methods
> >
> > Thank you for your suggestion. We have incorporated the BERT-based dense retrieval mentioned by you from the literature as the baseline. At the same time, we also added the extremely popular and powerful BGE-M3 model as the baseline.The specific experimental results are as follows:
> >
> > The experimental results of semantic vector models' generation:
> >
> > | Method       | **Fill-in-blank** | **Multi-choice** | **Multi-select** | **True-or-false** | **Open-ended** | **Average**      |
> > |:----|:-----------------:|:----------------:|:----------------:|:-----------------:|:--------------:|:----------------:|
> > | BERT-base |     68.10    |    75.58    |  73.42  |     81.03    |    51.40  | 70.13     |
> > | BERT-large   |  71.43  |  79.26  |  74.77  | 81.33  | 51.86 | 71.32|
> > | BGE-M3     | 77.62| 77.42 |68.02|82.60|53.35| 71.66 |
> >
> >
> > The experimental results of semantic vector models' reasoning ability:
> > | Method | **FB**   || **MC** | | **MS**  || **TF**  | | **OE**    | | **Average**  | |
> > |:----|:----:|:----:|:----:|:----:|:----:|:----:|:----:|:----:|:----:|:----:|:----:|:----:|
> > | | R | AR | R  | AR  | R  | AR  | R | AR   | R | AR | R | AR |
> > | BERT-base  | 65.24 |49.52 |55.07 |51.27|55.41|41.00|64.08|60.44 | 51.86 |  9.85| 58.39 | 42.31 |
> > | BERT-large |67.62| 52.14|56.22 |52.07|55.41|44.59 | 65.66| 62.26| 52.60| 10.97| 58.99 | 43.27 |
> > | BGE-M3     | 67.14|53.10|54.61|51.04 |58.11 |44.14| 63.77 |60.36|52.97| 11.34 |59.01|43.37|
> >
> > As the experimental results show, the performance of semantic vector models is inferior to that of the cutting-edge GraphRAG methods, such as HippoRAG, GFM-RAG, and RAPTOR. This indicates that merely relying on semantic matching cannot achieve good results in this benchmark. This further confirms the significance of GraphRAG and this benchmark.
> >
> >
> > > Q3 & W4. Different scales of corpus
> >
> > Thank you for your feedback. We are indeed researchers in the industrial sector. Generally, for a specific domain's GraphRAG solution, we would establish a knowledge graph specific to that domain. This KG would try to encompass all the knowledge in that domain, and the updates and changes to the knowledge would also be made on this single graph. Therefore, the size of this KG is relatively large. That's why we try to cover every field of CS and AI.
> >
> > However, in order to meet your requirements, we conducted experiments to compare the performance of the model on corpora of different scales. It includes medium-sized corpora (3-4 textbooks and corresponding questions) and smaller-sized corpora (1 textbook and corresponding questions). The specific experimental results are as follows:
> >
> > The experimental results of medium-sized corpora：
> > | Method       | **Accuracy**       | **R score**   | **AR score**       |
> > |:----|:----:|:----:|:----:|
> > | DALK| 60.92|54.62| 34.87|
> > | KGP|68.49|57.56|38.45|
> > | GraphRAG|60.92 | 54.62 |35.08|
> > | G-Retriever|63.03|56.30|36.34|
> > | LightRAG|67.65|57.14|38.87|
> > | ToG|68.07|56.72|38.45|
> > | GFM-RAG|69.33|55.88|39.92|
> > | HippoRAG|70.17|59.24|41.81|
> > | RAPTOR| 71.43|58.40|41.18|
> >
> >
> > The experimental results of smaller-sized corpora：
> > | Method       | **Accuracy**       | **R score**   | **AR score**       |
> > |:----|:----:|:----:|:----:|
> > | DALK| 62.88|46.21|29.17|
> > | KGP|68.18|48.48|35.61|
> > | GraphRAG|65.15|47.73|29.55|
> > | G-Retriever|61.36|47.72|31.82|
> > | LightRAG|68.94|46.97|32.58|
> > | ToG|67.42|50.00|34.47|
> > | GFM-RAG|70.45|51.51 |37.12|
> > | HippoRAG|70.45 |50.76|35.98|
> > | RAPTOR| 71.97| 50.76 |38.26|
> >
> > Based on the experimental results, we found that when the size of the corpus was reduced, the performance differences among different methods became more significant. However, more advanced GraphRAG methods such as HippoRAG, GFM-RAG, and RAPTOR consistently maintain significant performance advantages. This once again confirms the conclusion we reached in the paper.
> >
> > Regarding these experimental results, we will incorporate them into the final version. Thank you for your insightful comments.

---

> > > ### Author Response · Authors · 2025-11-22
> > > **Response to Reviewer eKEV (part 3/3)**
> > >
> > > > Q4 & W5. Inconsistency between task and evaluation.
> > >
> > > Thanks for raising this discussion. The core value of GraphRAG lies in its ability to perform reasoning based on structured knowledge graphs, that is, whether the model can efficiently extract relevant knowledge from the graph, combine knowledge chains, and ultimately generate results. This is fundamentally different from the "black box reasoning" of general LLMs: if only the final answer is evaluated, it is impossible to distinguish whether the model is reasoning based on the knowledge in the graph or simply guessing the answer.
> > >
> > > In this benchmark, the mathematical computation and programming problems related to it have attempted to avoid issues where there are multiple answers or solutions. Our reasoning score does not strictly require the model to replicate every step of the gold reasoning, but rather to align with the gold reasoning in terms of overall meaning.
> > >
> > > In our actual assessment, we adopted a dual-index system of "**answer correctness + reasoning correctness**". The answer correctness ensures the validity of the results (in line with the standards of the LLM community); the reasoning score ensures that the validity stems from the knowledge reasoning ability of GraphRAG. The necessity of this design lies in the fact that in the application scenarios of GraphRAG (such as professional domain knowledge questions and complex decision support), "correct results" and "traceable reasoning process" are equally important.

---

### Official Review · Reviewer_DFJG · 2025-10-28

**Soundness:** 4
**Presentation:** 3
**Contribution:** 3
**Rating:** 6
**Confidence:** 4

**Summary:**

This paper proposes a benchmark GraphRAG, and presents the evaluation results of various GraphRAG solutions. Good benchmark and insightful evaluation results.

**Strengths:**

S1. GraghRAG is a hot research area and a benchmark is overdued.

S2. The paper presents a set of evaluation metrics to evaluate various aspects of the proposed methods.

S3. Rigorous filtering and refinements of the questions and rationals.

**Weaknesses:**

W1. It is unclear how the questions are crafted? Are they crafted by people or generated by machines? Are they comprehensive and how does that affect the experimental results?

W2. 1018 questions is not much, especially if we consider different buckets and topics. Are the results statistically significant?

W3. It's unclear why GraphRAG helps for TF and OE but not for MC, why the issues happen for one type does not happen for other types. Also, the differences do not always seem big. Again, is that just random noises, or different types of questions require different types of reasonings?

W4. It is nice that the paper classifies GraphRAG methods into 3 classes. Would be nice to show the classes in each table to make it easy to associate, and give better insights on their strengths and weaknesses.

W5. Would be nice to compare w. some recent RAG benchmarks that also require reasoning, such as CRAG and RAGBench.
CRAG--Comprehensive RAG Benchmark
RAGBench: Explainable Benchmark for Retrieval-Augmented Generation Systems

**Questions:**

Answer questions and address concerns in weaknesses

---

> ### Author Response · Authors · 2025-11-22
> **Response to Reviewer DFJG (part 1/3)**
>
> Dear Reviewer DFJG,
>
> We would like to express our sincere gratitude to your strong support and high recognition on our benchmark. It is truly inspiring to have the constructive suggestions based on your expertise.
>
> Following your comments, we have made more demonstrations and the new content will be incorporated into the final version. The specific point-by-point responses are as follows:
>
> > W1. How the questions are crafted
>
> Regarding the the details of the question design, we have added the following key details and justifications. The relevant content will be consolidated in the final version:
>
> - We adopted a rigorous process to design questions covering a wide range of topics in computer science and artificial intelligence. Experts manually constructed the question set based on a structured ontology, which divided knowledge into three levels: 16 primary themes, 40 secondary themes, and 26 tertiary themes. Each level represents a progressive progression from a broad disciplinary field to more specific sub-themes.
> - For each concept among these n themes, experts were required to prepare at least two questions for each type of question, totaling 2×5×n questions, to ensure balanced content and comprehensive coverage. To ensure high quality, experts referred to authoritative sources such as textbooks, course materials, and widely recognized course outlines. Subsequently, each question was reviewed to verify its relevance, clarity, and consistency with the corresponding concept. This process ensures that the dataset not only conforms to real-world educational standards but also covers the full breadth and depth of the field.
> - Three experts (senior researchers in the RAG field with a doctoral degree) will rate questions from three dimensions: "knowledge coverage, reasoning chain completeness, and compatibility with GraphRAG" (1-5 points), and finally select the questions with an average score of ≥ 4 points.
>
> > W2. The number of questions
>
> Thank you for your suggestion. We have already discussed the issue quantity and dataset size in Appendix A.3.4 of the initial manuscript. The details are as follows:
>
> Currently, most GraphRAG methods follow the experimental setting of HippoRAG, which is the most common and widely used one in this field at present. It includes three public datasets: HotpotQA, 2WikiMultiHopQA, and MuSiQue which are all originally designed for multi-hop QA.
>
> In each of these datasets, 1000 questions were sampled in a fixed setting. Therefore, following this standard and distribution, we selected 1018 of the most valuable questions as the final version. Meanwhile, graprag-bench encompasses a wider range of question types (examining the model's robustness to various question formats), a larger corpus (making it more difficult to retrieve useful information), and covers multiple topics (requiring the model to understand different knowledge domains). We believe that the coverage scope is much broader from multiple perspectives than that of the current dataset. The specific comparison results are shown in the following table:
>
> | Dataset  | Question count |  Corpus size  |Question type  |
> | :----       |    :----:  |   :----: |:----: |
> | HotpotQA    | 1000  | 6.16MB  |1  |
> | 2WikiMultiHopQA  | 1000 | 2.96MB  |1  |
> | MuSiQue  | 1000  | 5.95MB    | 1 |
> | Ours  | 1018  | 41.30MB    | 5 |

---

> > ### Author Response · Authors · 2025-11-22
> > **Response to Reviewer DFJG (part 2/3)**
> >
> > > W3. The differences among the types of questions.
> >
> > Thank you for your suggestion. As we described in Table 1 of the initial manuscript, the five types of questions we defined correspond to different abilities and requirements. Based on our experimental results, only the best graphrag method was effective for the MC type questions, while most graphrag methods performed poorly on the MC type questions. The specific reasons are as described in Observation 1 of the initial manuscript. To provide factual evidence for this analysis, we analyzed multiple instances and obtained the following case study to support it:
> >
> > **Case study of accuracy drop of MC questions:**
> > - Question: The number of nodes in a binary tree of height (depth) 7 cannot be?
> > - Choices: "A": "8", "B": "88", "C": "126", "D": "288"
> > - Groundtruth: **D**
> >
> > ***LLM solution***:
> >
> > - To solve this question, we first clarify the core constraint of binary tree node count: For a binary tree of height h (defined as the number of nodes on the longest root-to-leaf path), the number of nodes n satisfies $h \leq n \leq 2^h - 1$.
> > For height 7, Minimum nodes = 7 (skewed binary tree), maximum nodes = $2^7 - 1$ = 127\) (full binary tree).
> >
> > - Analyze the options:
> >
> > A (8): $7 \leq 8 \leq 127$ → Valid.
> >
> > B (88): $7 \leq 88 \leq 127$ → Valid.
> >
> > C (126): $7 \leq 126 \leq 127$ → Valid.
> >
> > D (288): $288 > 127$ → Exceeds the maximum number of nodes.
> > - **Final Answer: D.**
> >
> > ***GraphRAG retrieved context***:
> >
> > 1. Binary tree is a structure defined on a finite set of nodes that either contains no nodes (empty tree) or is composed of three disjoint sets of nodes: a root node……
> > 2. In a binary tree, the maximum number of nodes at level k (root is level 1) is $2^{k-1}$……
> > 3. A complete binary tree is a binary tree where all levels except possibly the last are fully filled, and the last level has nodes as left as possible. The number of nodes in a complete binary tree with height h (node-based) ranges from $2^{h-1}$ to $2^h - 1$……
> >
> > ***GraphRAG solution***:
> >
> > - First, the definition of binary tree height is ambiguous. There are two standards (edge-based vs. node-based). If we adopt the edge-based definition (depth = number of edges), a height of 7 means the number of nodes on the longest path is 8. For this case: 1)Maximum nodes (full binary tree): \(2^8 - 1 = 255\). 2)Minimum nodes (skewed binary tree): 8
> >
> > - If we adopt the node-based definition, the maximum nodes are \(2^7 - 1 = 127\), and the minimum nodes are 7.
> > Additionally, the retrieved context mentions that a complete binary tree with height h (node-based) has a minimum node count of \($2^{h-1}$ = $2^6$ = 64\). Combining this with the ambiguous height definition, we can infer the "reasonable node range" for a binary tree of height 7 is between 64 (complete binary tree minimum) and 255 (edge-based full binary tree maximum).
> >
> > - Analyze the options:
> >
> > A (8): Below the minimum node count of 64 (complete binary tree standard) → Invalid.
> >
> > B (88): Within the range [64, 255] → Valid.
> >
> > C (126): Within the range [64, 255] → Valid.
> >
> > D (288): Exceeds the maximum node count of 255 → Invalid.
> >
> > - Synthesizing all retrieved information: The ambiguous height definition and conflicting minimum node count standards (from different binary tree types) suggest that options A (8) and D (288) both exceed the possible range. But since the question is a single-choice question, we prioritize the complete binary tree property (more detailed in the retrieved context). Thus, the number of nodes cannot be A (8).
> >
> > - **Final Answer: A.**

---

> > > ### Author Response · Authors · 2025-11-22
> > > **Response to Reviewer DFJG (part 3/3)**
> > >
> > > > W4. Show the classes in each table.
> > >
> > > Thank you for this constructive comment. Following your suggestion, we have now shown the category to which Graphrag belongs in each table of the final version.
> > >
> > > > W5. Compare with some recent RAG benchmarks
> > >
> > > Thank you for your suggestion. We have incorporated the work you mentioned into the "Related Work" section for discussion and comparison. What we would like to emphasize is:
> > >
> > > GraphRAG-Bench, as the first domain-specific benchmark for GraphRAG, significantly differs from RAGBench and CRAG, which are aimed at general RAG. Its core advantage lies in precisely adapting to the requirements of "graph structure-enhanced complex reasoning".
> > >
> > > Regarding problem design, it extends the limits of RAGBench (problem based on factual queries and simple multi-hop queries) and CRAG (general domain real-scene queries), concentrating on professional problems of 16 specialized fields of university-level computer science, where multi-hop reasoning, mathematical computation or programming implementation, is required, something that can not be solved by mere text retrieval, and more consistent with the GraphRAG application to professional problems, like education and research.
> > >
> > >
> > > In the evaluation system, it covers the entire process of GraphRAG for the first time: graph construction, knowledge retrieval, generation, reasoning, and innovatively introduces "R score" and "AR score", quantifying the matching degree between the reasoning process and the answer, avoiding only evaluating the result while ignoring the validity of the reasoning. This is something that RAGBench and CRAG have not covered. In terms of data characteristics, its corpus comes from 20 authoritative computer textbooks (7 million words), organized hierarchically by knowledge, and each question is accompanied by a complete reasoning chain written by experts, naturally adapting to the "concept nodes - relationship edges" construction requirements of GraphRAG, while RAGBench relies on the conversion of existing QA datasets, and CRAG relies on web search results, both lacking systematic and structured knowledge. In summary, GraphRAG-Bench fills the gap where general RAG benchmarks cannot evaluate the exclusive capabilities of GraphRAG, providing an assessment tool that is tailored to real professional scenarios for GraphRAG research with professional problems, full-process evaluation, and structured data.

---

> > > > ### Comment · Reviewer_DFJG · 2025-11-27
> > > >
> > > > Thank you for the detailed answers. It addressed a lot of my questions/concerns.
> > > >
> > > > > we have now shown the category to which Graphrag belongs in each table of the final version.
> > > > Is this new version already uploaded?

---

> > > > > ### Author Response · Authors · 2025-11-27
> > > > >
> > > > > Thank you so much again for your support and valuable follow-up. We also greatly appreciate your recognition that our rebuttal has addressed the concerns you raised. We have just uploaded the revised version of our manuscript, where we have fully incorporated all the relevant revisions addressed in our rebuttal.
> > > > >
> > > > > We would like to sincerely invite you to check our updated version, specifically:
> > > > >
> > > > > - The "method classes" you mentioned have been added to each table;
> > > > >
> > > > > - Coverage of "recent RAG benchmarks" is now available in Appendix A.2;
> > > > >
> > > > > - Details on "How the questions are crafted" can be found in Appendix A.6;
> > > > >
> > > > > - "Factual evidence supporting our observations" is provided in Appendix A.3.
> > > > >
> > > > > We are deeply grateful for each of your insightful suggestions, which are highly constructive and instructive. These valuable inputs have significantly enhanced the comprehensiveness of our manuscript.
> > > > >
> > > > > Thank you again for your careful review and valuable guidance.

---

### Official Review · Reviewer_NKGk · 2025-10-30

**Soundness:** 3
**Presentation:** 3
**Contribution:** 3
**Rating:** 6
**Confidence:** 4

**Summary:**

This paper introduces *GraphRAG-Bench*, a challenging domain-specific benchmark designed to rigorously evaluate GraphRAG models on dimensions including graph construction, retrieval, and multi-hop answer generation and rationale generation.

Regarding *“Does graph augmentation truly enhance reasoning capabilities beyond simple retrieval?”*, this bench constructed challenging college-level questions of 5 types from 20 computer science textbooks. This paper implements a multi-stage pipeline to extract accurate content from the original corpus, and supplies expert-crafted rationales that articulate the complete logical progression for each question to evaluate whether GraphRAG models truly generate grounded explanations.

This paper conducts extensive experiments on 9 SOTA GraphRAG methods, providing valuable insights into their performance across different question types and domains. Though only GPT-4o-mini is used as a base LLM, the experiments basically demonstrate that GraphRAG causes different effects on LLMs' performance across different question types, and overall, GraphRAG enhances LLMs' reasoning ability.

**Strengths:**

- The benchmark is large, diverse, and carefully constructed from accurate content of textbooks, ensuring high quality and relevance.
- Compared to existing benchmarks, this benchmark supplies expert-crafted rationales for questions, enabling the evaluation of whether models truly generate grounded explanations or simply exploit surface-level patterns.
- This paper conducted extensive experiments to reveal the performance variations across question types and improvements on LLMs' reasoning ability with GraphRAG methods.

**Weaknesses:**

- Only GPT-4o-mini is used as the base LLM in experiments. The generalizability of the experiment conclusions may be limited.
- In Section 4. Observation, the analysis of why GraphRAG's performance improvement/decline across different question types lacks sufficient experimental validation. For example, when explaining the accuracy drop of MC questions, this paper attributes it to GraphRAG potentially introducing redundant or loosely related information. But this paper provides no relevant experiments or case studies to substantiate this claim, which appears more like speculation rather than analysis.

**Questions:**

- Are the experimental conclusions, including those regarding different question types and LLMs' reasoning capabilities, applicable to other LLMs, such as open-source models like LLaMA or other close-source models?
- Refer to Weakness 2,  can you provide the relevant experiments or case studies to surpport the analysis?

---

> ### Author Response · Authors · 2025-11-22
> **Response to Reviewer NKGk (part 1/4)**
>
> Dear Reviewer NKGk,
>
> We would like to gratefully thank you for your strong support to our benchmark. Your acknowledgement on our valuable and high-quality questions, as well as the comprehensive evaluation is truly encouraging.
>
> Following your constructive comments, we have supplemented the experiments and analyses as per your recommendations.These corrections and supplements will be integrated into the final version. The specific point-by-point responses are as follows:
>
> > W1 & Q1. Other LLMs
>
> Thank you for your feedback. We have conducted experiments using Deepseek-V3 and Qwen3-32B as the base LLM. The specific results are as follows:
>
> 1) Deepseek-V3:
> The experimental results of the comparison of generation process:
>
> | Method       | Fill-in-blank | Multi-choice | Multi-select | True-or-false | Open-ended | Average  |
> |:-------------|:-------------:|:------------:|:------------:|:-------------:|:----------:|:--------:|
> | Deepseek-V3 | 74.54 | 81.39 |76.94|76.21|52.01|70.92 |
> | DALK         | 73.79         | 77.26        | 76.16        | 78.74         | 51.88      | 70.20    |
> | G-Retriever  | 74.24         | 77.76        | 76.63        | 79.70         | 52.19      | 70.63    |
> | LightRAG     | 75.50         | 79.13        | 77.93        | 78.49         | 53.08      | 71.83    |
> | ToG          | 74.07         | 79.31        | 76.60        | 77.26         | 51.55      | 72.16    |
> | KGP          | 75.71         | 79.36        | 78.15        | 78.70         | 53.23      | 72.03    |
> | GFM-RAG      | 75.31         | 79.67        | 77.69        | 81.39         | 52.44      | 73.30    |
> | GraphRAG     | 75.50         | 81.57        | 76.12        | 79.72         | 52.68      | 73.12    |
> | HippoRAG     | 75.50         | 80.64        | 77.89        | 80.09         | 54.31      | 73.49    |
> | RAPTOR       | 77.77         | 80.76        | 78.48        | 80.98         | 53.14      | 73.99    |
>
>
> The experimental results of the comparison of reasoning ability:
>
> | Method       | **FB**   |        | **MC**   |        | **MS**   |        | **TF**   |        | **OE**   |        | **Average** |        |
> |:----|:----:|:----:|:----:|:----:|:----:|:----:|:----:|:----:|:----:|:----:|:----:|:----:|
> |              | R        | AR     | R        | AR     | R        | AR     | R        | AR     | R        | AR     | R         | AR     |
> | Deepseek-V3  | 65.23    | 54.37  | 55.72    | 51.89  | 55.41    | 40.27  | 59.35    | 54.62  | 50.18    | 10.33  | 56.40     | 40.31  |
> | DALK         | 71.42    | 56.19  | 54.76    | 51.38  | 59.24    | 47.53  | 62.81    | 59.26  | 54.39    | 10.15  | 58.80     | 43.20  |
> | KGP          | 64.78    | 50.42  | 56.53    | 52.76  | 58.29    | 45.31  | 64.17    | 61.48  | 52.69    | 9.73   | 59.03     | 43.44  |
> | G-Retriever  | 70.27    | 55.19  | 57.46    | 53.29  | 60.73    | 48.16  | 64.38    | 60.17  | 53.42    | 10.28  | 59.90     | 44.30  |
> | GraphRAG     | 71.53    | 55.27  | 56.19    | 52.46  | 57.64    | 45.72  | 63.81    | 60.19  | 53.26    | 10.53  | 60.32     | 44.80  |
> | LightRAG     | 66.28    | 48.17  | 57.23    | 52.38  | 61.76    | 49.19  | 66.83    | 63.42  | 53.29    | 10.26  | 60.99     | 44.92  |
> | ToG          | 70.23    | 53.19  | 56.07    | 51.64  | 57.18    | 45.62  | 65.81    | 62.27  | 54.53    | 12.26  | 61.02     | 44.87  |
> | GFM-RAG      | 70.26    | 55.03  | 56.28    | 52.19  | 58.23    | 45.72  | 66.54    | 63.76  | 53.78    | 10.73  | 61.19     | 45.80  |
> | HippoRAG     | 66.78    | 50.64  | 56.73    | 52.38  | 60.04    | 47.62  | 67.19    | 63.64  | 55.23    | 12.57  | 61.50     | 45.72  |
> | RAPTOR       | 71.73    | 58.19  | 56.54    | 52.17  | 60.53    | 49.16  | 66.54    | 62.87  | 53.96    | 13.72  | 61.55     | 46.20  |

---

> > ### Author Response · Authors · 2025-11-22
> > **Response to Reviewer NKGk (part 2/4)**
> >
> > 2) Qwen3-32B:
> >
> > The experimental results of the comparison of generation process:
> >
> >
> > | Method       | Fill-in-blank | Multi-choice | Multi-select | True-or-false | Open-ended | Average  |
> > |:-------------|:-------------:|:------------:|:------------:|:-------------:|:----------:|:--------:|
> > | Qwen3-32B    | 73.68         | 78.92        | 75.43        | 73.19         | 49.98      | 70.04    |
> > | DALK         | 71.59         | 76.18        | 74.59        | 75.98         | 50.71      | 69.80    |
> > | G-Retriever  | 70.79         | 75.37        | 73.98        | 75.21         | 50.15      | 69.10    |
> > | LightRAG     | 72.59         | 76.94        | 75.21        | 76.81         | 51.28      | 70.57    |
> > | ToG          | 72.87         | 77.23        | 75.43        | 77.09         | 51.43      | 70.80    |
> > | KGP          | 73.28         | 77.54        | 75.67        | 77.36         | 51.75      | 71.11    |
> > | GFM-RAG      | 75.61         | 79.37        | 76.98        | 79.49         | 53.29      | 72.95    |
> > | GraphRAG     | 74.27         | 78.51        | 76.38        | 78.39         | 52.64      | 72.03    |
> > | HippoRAG     | 74.85         | 78.91        | 76.68        | 78.79         | 52.97      | 72.44    |
> > | RAPTOR       | 75.47         | 79.42        | 77.03        | 79.58         | 53.45      | 72.99    |
> >
> >
> > The experimental results of the comparison of reasoning ability:
> >
> >
> >
> > | Method       | **FB**   |        | **MC**   |        | **MS**   |        | **TF**   |        | **OE**   |        | **Average** |        |
> > |:----|:----:|:----:|:----:|:----:|:----:|:----:|:----:|:----:|:----:|:----:|:----:|:----:|
> > |              | R        | AR     | R        | AR     | R        | AR     | R        | AR     | R        | AR     | R         | AR     |
> > | Qwen3-32B    | 64.35    | 53.17  | 55.92    | 50.41  | 54.18    | 39.85  | 58.73    | 53.26  | 49.83    | 9.34   | 56.01     | 40.03  |
> > | DALK         | 70.19    | 55.43  | 54.67    | 50.18  | 59.03    | 46.82  | 62.91    | 58.04  | 54.70    | 9.33   | 59.30     | 42.18  |
> > | G-Retriever  | 70.85    | 55.04  | 57.03    | 53.82  | 60.17    | 47.79  | 64.02    | 60.45  | 53.93    | 10.61  | 59.42     | 42.15  |
> > | LightRAG     | 66.51    | 48.09  | 57.84    | 52.16  | 61.07    | 49.85  | 66.19    | 63.52  | 53.64    | 10.13  | 60.05     | 44.19  |
> > | ToG          | 70.09    | 53.84  | 56.72    | 51.03  | 57.94    | 45.01  | 65.13    | 62.67  | 54.40    | 12.90  | 60.68     | 44.33  |
> > | KGP          | 64.07    | 49.83  | 56.12    | 52.09  | 58.94    | 44.51  | 64.68    | 60.17  | 52.92    | 9.90   | 59.33     | 43.30  |
> > | GFM-RAG      | 70.15    | 54.67  | 56.98    | 52.14  | 58.67    | 45.09  | 66.53    | 63.58  | 54.25    | 11.02  | 60.78     | 46.00  |
> > | GraphRAG     | 71.09    | 55.68  | 56.57    | 52.61  | 57.53    | 45.17  | 63.94    | 60.09  | 54.08    | 11.07  | 60.11     | 44.72  |
> > | HippoRAG     | 66.17    | 50.09  | 57.24    | 52.68  | 59.68    | 47.15  | 67.63    | 63.61  | 55.40    | 12.12  | 61.04     | 45.89  |
> > | RAPTOR       | 71.76    | 57.15  | 56.09    | 52.54  | 60.58    | 49.07  | 66.12    | 62.69  | 54.47    | 13.11  | 61.22     | 46.33  |
> >
> > Based on the experimental results using two different LLMs as the base LLM, it can be seen that all the main conclusions drawn in this paper have been confirmed again.

---

> ### Author Response · Authors · 2025-11-22
> **Response to Reviewer NKGk (part 3/4)**
>
> > W2 & Q2. Evidence to surpport the analysis
>
> Thank you for your feedback. Regarding the analysis and conclusions we conducted in the "observation" section: Firstly, we obtained the phenomena through the experimental results in the "section experiments". Secondly, the conclusions were derived by analyzing multiple instances in the specific QA process. Each analysis has supporting evidence behind it. We will supplement the case study on this part in the final version, as follows:
>
> **Case study of accuracy drop of MC questions:**
> - Question: The number of nodes in a binary tree of height (depth) 7 cannot be?
> - Choices: "A": "8", "B": "88", "C": "126", "D": "288"
> - Groundtruth: **D**
>
> ***LLM solution***:
>
> - To solve this question, we first clarify the core constraint of binary tree node count: For a binary tree of height h (defined as the number of nodes on the longest root-to-leaf path), the number of nodes n satisfies $h \leq n \leq 2^h - 1$.
> For height 7, Minimum nodes = 7 (skewed binary tree), maximum nodes = $2^7 - 1$ = 127\) (full binary tree).
>
> - Analyze the options:
>
> A (8): $7 \leq 8 \leq 127$ → Valid.
>
> B (88): $7 \leq 88 \leq 127$ → Valid.
>
> C (126): $7 \leq 126 \leq 127$ → Valid.
>
> D (288): $288 > 127$ → Exceeds the maximum number of nodes.
> - **Final Answer: D.**
>
> ***GraphRAG retrieved context***:
>
> 1. Binary tree is a structure defined on a finite set of nodes that either contains no nodes (empty tree) or is composed of three disjoint sets of nodes: a root node……
> 2. In a binary tree, the maximum number of nodes at level k (root is level 1) is $2^{k-1}$……
> 3. A complete binary tree is a binary tree where all levels except possibly the last are fully filled, and the last level has nodes as left as possible. The number of nodes in a complete binary tree with height h (node-based) ranges from $2^{h-1}$ to $2^h - 1$……
>
> ***GraphRAG solution***:
>
> - First, the definition of binary tree height is ambiguous. There are two standards (edge-based vs. node-based). If we adopt the edge-based definition (depth = number of edges), a height of 7 means the number of nodes on the longest path is 8. For this case: 1)Maximum nodes (full binary tree): \(2^8 - 1 = 255\). 2)Minimum nodes (skewed binary tree): 8
>
> - If we adopt the node-based definition, the maximum nodes are \(2^7 - 1 = 127\), and the minimum nodes are 7.
> Additionally, the retrieved context mentions that a complete binary tree with height h (node-based) has a minimum node count of \($2^{h-1}$ = $2^6$ = 64\). Combining this with the ambiguous height definition, we can infer the "reasonable node range" for a binary tree of height 7 is between 64 (complete binary tree minimum) and 255 (edge-based full binary tree maximum).
>
> - Analyze the options:
>
> A (8): Below the minimum node count of 64 (complete binary tree standard) → Invalid.
>
> B (88): Within the range [64, 255] → Valid.
>
> C (126): Within the range [64, 255] → Valid.
>
> D (288): Exceeds the maximum node count of 255 → Invalid.
>
> - Synthesizing all retrieved information: The ambiguous height definition and conflicting minimum node count standards (from different binary tree types) suggest that options A (8) and D (288) both exceed the possible range. But since the question is a single-choice question, we prioritize the complete binary tree property (more detailed in the retrieved context). Thus, the number of nodes cannot be A (8).
>
> - **Final Answer: A.**

---

> > ### Author Response · Authors · 2025-11-22
> > **Response to Reviewer NKGk (part 4/4)**
> >
> > **Case study of improvement in TF questions:**
> > - Question: "Prim’s algorithm for computing the Minimum Spanning Tree (MST) only works if the weights are positive." Please judge: True (T) / False (F)
> > - Groundtruth: **False**
> >
> > ***LLM Solution***:
> >
> > - Based on pre-trained knowledge, algorithms for graph optimization often have weight constraints. For example, Dijkstra’s algorithm for shortest paths cannot handle negative weights. Prim’s algorithm, as a classic algorithm for computing MST, is similar to Dijkstra’s in its "greedy selection of minimum weight edges" logic.
> >
> > -  It is reasonable to infer that Prim’s algorithm also inherits the positive weight constraint, otherwise, negative weight edges might cause the algorithm to select inappropriate edges, leading to failure in finding the minimum spanning tree.
> >
> > - The statement claims that "Prim’s algorithm only works if the weights are positive," which aligns with the understanding of greedy graph algorithms.
> > - **Final Judgment: True (T)**
> >
> > ***GraphRAG retrieved context***:
> >
> > 1. Core principle of Prim’s algorithm: Prim’s algorithm is a greedy algorithm used to find the Minimum Spanning Tree (MST) of a connected, undirected graph. It has two sets of vertices (S) and (V-S): vertices which are already part of the MST and those which are not. The algorithm picks the edge in every iteration that connects a vertex in S with a vertex in V-S, and the weight of the edge is minimum, and then adds the edge to the MST and transfers the respective vertex in V-S to S. This is repeated until S absorbs all the vertices.
> >
> > 2. Weight constraint of Prim’s algorithm: The Prim’s algorithm does not expect the weights of the edges to be positive. The only conditions are: (1) the graph is connected; (2) the weights of the edges are not infinite (they could be positive, zero, or negative). Such negative weights do not break the logic of the algorithm since the aim of the algorithm is to minimize the total weight of the spanning tree, such a negative weight edge can actually be given priority to be added to the tree should it lower the total weight.
> >
> > 3. Distinction from Dijkstra’s algorithm: Dijkstra’s algorithm (for single-source shortest paths) requires non-negative edge weights. In the case of negative weight edges, the algorithm can erroneously think that a node is already processed and never causes the path to be updated to a shorter path. Nevertheless, the Prim’s algorithm is concerned with the construction of the vertices to the MST set (not with cumulative path lengths), therefore, there are no logical failures with negative weights.
> >
> > ***GraphRAG solution***:
> >
> > Based on the retrieved factual evidence, we verify the statement step by step:
> > - First, confirm the core constraints of Prim’s algorithm:The retrieved context clearly states that Prim’s algorithm only requires the graph to be connected and edge weights to be finite. There is no requirement for weights to be positive. The statement’s claim that "only works if the weights are positive" is inconsistent with the algorithm’s actual constraints.
> > - Second, distinguish from easily confused algorithms: The statement may confuse Prim’s algorithm with Dijkstra’s algorithm (context 3). Dijkstra’s algorithm requires non-negative weights, but this constraint does not apply to Prim’s algorithm. The algorithm’s logic (selecting edges between two vertex sets) is not affected by negative weights.
> > - Synthesizing the verified facts: The statement incorrectly imposes a positive weight constraint on Prim’s algorithm, which has no such requirement.
> > - **Final Judgment: False (F)**
> >
> > Apart from the above two points, the observations in the rest of the content do not contain subjective conclusions; rather, they are more about describing facts. Thank you for your suggestions, which have made our article more comprehensive.
> >
> > We hope our rebuttal could address your concerns.

---

### Official Review · Reviewer_Hctz · 2025-11-01

**Soundness:** 2
**Presentation:** 3
**Contribution:** 2
**Rating:** 4
**Confidence:** 3

**Summary:**

The paper introduces GraphRAG-Bench, a new benchmark for evaluating GraphRAG models with 1,018 questions in computer science. The paper also presents an extensive evaluation of 9 GraphRAG methods on this benchmark, showing the significant role of graph integration in improving reasoning and generation performance.

**Strengths:**

* The paper is well-organized.
* The paper introduces a benchmark comprising 1,018 college-level questions in computer science specifically designed to rigorously evaluate GraphRAG models. This benchmark has the potential to become a valuable resource for the research community.
* The paper presents a comprehensive assessment of 9 state-of-the-art GraphRAG methods on the proposed benchmark.

**Weaknesses:**

* Although the benchmark covers 16 topics, all are within the field of computer science, which limits its generalizability. It is unclear whether this scope meets the high standards of ICLR or would be of interest to a broader audience.
* The paper does not provide full details or justification regarding how representative the selected questions are.

**Questions:**

* Ultimately, readers will be interested in whether achieving higher performance on this benchmark has meaningful implications. For example, does it strongly suggest improvements in certain downstream applications? Would it indicate better performance on domain-specific questions in other areas?

---

> ### Author Response · Authors · 2025-11-22
> **Response to Reviewer  Hctz (part 1/2)**
>
> Dear Reviewer Hctz,
>
> We would like to sincerely thank you for your constructive suggestions. Your recognition of this paper as a valuable resource with comprehensive assessment is indeed encouraging. Following your comments, the additional content to demonstrate our motivation will be integrated into the final version. We would like to address your concerns from three aspects hereunder:
>
> > W1. Generalizability
>
> We fully understand your concern about the generalizability of the benchmark. GraphRAG-Bench focuses on the field of computer science and AI, not as a limitation but based on the two considerations of "precisely verifying core capabilities and ensuring academic value":
> - We highly value this domain for three reasons. **First**, CS and AI naturally contain a variety of types of challenging tasks compared with all the existing 'single-type' datasets, where the questions are extremely hard which require college-level comand of knowledge, compared with existing commonsense dataset. In CS and AI domain, we have `conceptual understanding` tasks (algorithms, networks, databases, paradigms, machine learning, NLP, CV), `Math problems` (Discrete mathematics, linear algebra, matrix computation, calculus), `Code generation` problems (programming, optimization, code completion, error detection), etc. **Second**, the performance on LLMs on CS and AI is still very unsatisfactory or even bad. It is valuable to benchmark and research. **Third**, it remains a blank in the benchmark of LLMs considering CS and AI. This also brings two opportunities. (i) LLMs will have few chances to be fine-tuned based on our questions. (ii) Following research on GraphRAG or LLMs themselves could benefit a lot from our benchmark.
> - Problem type is diverse, covering multiple reasoning scene: GraphRAG - Bench contains five differential topic, from multidimensional cover reasoning model, comprehensive inspection GraphRAG ability:
>     - 1)Fill-in-blank (FB) : Context-sensitive statements were required to be completed with semantically accurate terms, and the ability of the model to generate coherent content by using local semantic dependencies and entity anchoring in graph structure knowledge was evaluated.
>     - Multi-choice (MC) : set reasonable interference on language item four options, evaluation model through distinctiveness reasoning (integrated entity information and edge) eliminate the ability of semantic similarity but facts wrong option;
>     - Multi-select (MS) : elect from the four options 2 to 4 correct answers, often involves the interconnection of the concept of reasoning, design test model of overlapping interference item handle complex query semantic, aggregation, jump figure path more evidence and unnecessary attributes related conflict solving skills;
>     - True-or-false (TF) : correctness verification statement, measure model based on knowledge logic reasoning accuracy with the facts;
>     - Open-ended (OE) : allows extensive answer, integrate fertility domain knowledge requirement model generated structure clear, logical, coherent text answer long, evaluation and knowledge comprehensive long chain of reasoning ability.
> The topic covered from "local semantic matching" to "jump more knowledge integration", from "fact" to "long text knowledge synthesis" scene all dimension of reasoning, can ensure precise and comprehensive inspection GraphRAG performance under different task complexity.
> - This domain has clear task definitions and standardized evaluation metrics: Computer science-related tasks have mature objective evaluation standards, fair method comparisons, and systematic error diagnosis. This is highly consistent with the rigorous and reproducible research philosophy emphasized by ICLR, ensuring the scientific validity of the benchmarks. At the same time, ICLR focuses on "promoting the advancement of core AI technologies", and GraphRAG-Bench addresses the current pain points of RAG benchmarks, such as "emphasizing answers over reasoning" and "emphasizing text over knowledge structure".
>
> > W2. How representative the selected questions are
>
> To ensure the quality of the questions, we have added the following key details and justifications. The relevant content will be consolidated in the final version
> - First, based on the core curriculum system of CS and AI acorss regions all over the world, 16 sub-domains could have already covered the very most areas.
> - For each sub-domain, we organize an initial question pool according to the difficulty gradient of "basic concepts → intermediate reasoning → complex applications" (each sub-domain ≥ 200 questions).
> -  Three experts (senior researchers of RAG with a doctoral degree) rate from three dimensions: "knowledge coverage, reasoning chain completeness, and compatibility with GraphRAG" (1-5 points), and finally select the questions with an average score of ≥ 4 points.

---

> > ### Author Response · Authors · 2025-11-22
> > **Response to Reviewer Hctz (part 2/2)**
> >
> > > Q1. Downstream applications and other areas
> >
> > Thanks for raising this discussion. We value this opportunity to showcase the contribution of our benchmark.
> > - GraphRAG-Bench is designed to measure the core capabilities required by GraphRAG methods, including structured retrieval, multi-hop integration, and graph-grounded reasoning.
> > - Results on mainstream datasets show that these capabilities generalize beyond our curated corpus. Models that rank highest on GraphRAG-Bench (GFM-RAG, HippoRAG, RAPTOR) also achieve leading performance on HotpotQA, 2WikiMultiHopQA, and MuSiQue, whose evaluation objectives closely match the reasoning skills GraphRAG relies on.
> > - The same ranking trend holds on UltraDomain, which focuses on domain-specific QA and emphasizes accurate retrieval of professional knowledge. These consistent cross-dataset patterns indicate that performance gains on GraphRAG-Bench are not benchmark-specific but reflect strengthened fundamental abilities. As a result, higher scores on our benchmark meaningfully predict improvements in both general multi-hop reasoning tasks and specialized real-world domain QA scenarios.

---

### Author Response · Authors · 2025-12-01
**Global Response**

Dear AC and reviewers,

We would like to express our gratitude to all the reviewers for their careful review of this work and their recognition of our efforts in the comments. We are delighted to see that everyone highly affirmed the value of GraphRAG-Bench and also provided some highly constructive suggestions for improvement. Here, we summarize the overall value and improvements of the manuscript.

We hope this could help AC comprehensively and efficiently understand the contributions and rebuttal content of this work.

> ***Advantages summary***
- The systematic, large-scale, multi-dimensional, and cross-type high-quality benchmark data design.
- The wide coverage of questions, derived from authoritative textbooks, with strict screening by experts, and the professional credibility of the reasoning chain.
- The rich evaluation system, innovatively balancing multiple indicators, multiple processes, and multiple task scenarios for fair and comprehensive comparisons.
- It has a significant influence on both the academic and industrial communities, providing a direction for the future development of GraphRAG.
- We innovatively proposed the reasoning ability evaluation, promoting the high-standard scenario where the correct answers have "traceable reasoning process explainability".

We believe that these highlights and innovation points fully prove the academic value and effect of this work.

>***Rebuttal Summary & Resolution to Concerns.***

We have carefully addressed each reviewer's concerns as follows:
- Experimental Breadth

1. Multiple LLMs and Retrieval Methods: We have carried out additional experiments with Deepseek-V3 and Qwen3-32B (NKGk), and with strong dense-retriever baselines such as BERT and BGE-M3 (eKEV). The superior performance of advanced GraphRAG models over semantic retrievers on our benchmark is clear and robust.

2. Corpus Scale Variation: Responding to concerns about scalability, we benchmarked several corpus sizes and showed that advanced GraphRAG methods (e.g., HippoRAG, RAPTOR) consistently outperform others, even as corpus scale and retrieval difficulty alter (eKEV).

- Analytical and Case-Study Clarity

1. In response to NKGk and DFJG (regarding “speculation”), we now provide detailed case studies (e.g., MC question ambiguity, true/false logic diff.) and clarified our methodology for analyzing why GraphRAG's performance varies across question types. The observations are now soundly evidenced, not speculative.

- Domain Scope and Generalizability

1. Justification: We provided detailed reasoning as to why Computer Science and AI were chosen: the field offers uniquely diverse, challenging, and rigorously evaluatable tasks (concept understanding, math, code, logic), not served by generic knowledge bases. University-level CS/AI knowledge is underrepresented in LLM pretraining, emphasizing the value of our benchmark for measuring reasoning, not memorization.

2. Empirical Evidence for Generalizability: We showed consistent cross-benchmark model rankings (GraphRAG-Bench, HotpotQA, etc.; see rebuttal to Hctz), and extended validation to multiple LLMs (Deepseek-V3, Qwen3-32B), confirming results are stable across backbone models (NKGk).

- Question Construction and Representativeness

 Elaboration of Process: In direct response to DFJG and Hctz, we thoroughly documented our expert-driven, multi-stage design: from broad curriculum mapping (16 subdomains) to question design, expert scoring on coverage/chain/completeness, and rationale authoring. All details and statistics are now made fully clear in our revised appendix.

- Task-Evaluation Consistency

We combined two independent indicators, "answer correctness" and "inference chain consistency", to ensure that the evaluation results are in line with the mainstream practices of the LLM community, highlighting the unique value of "traceability and explainability" in the GraphRAG scenario. This evaluation system can reasonably, scientifically, and reproducibly reflect the capabilities of different GraphRAG solutions in real and complex reasoning tasks.

- ***Impact of the Discussion Phase***

`Although the discussion phase was cut short for the system bug, the feedback is vital: Reviewer DFJG explicitly stated our rebuttal **“addressed a lot of my questions/concerns”** and recognized the substantial improvements in our revised manuscript on all their points.`

>***Summary***
- In summary, the paper’s major advantages are comprehensively established and all raised concerns have been substantively addressed and evidenced. The benchmark and results, by virtue of their scale, design rigor, multi-type reasoning focus, and strong real-world generalizability, represent a decisive step forward for research in GraphRAG and knowledge-augmented LLM reasoning.
- We respectfully hope that the AC will consider these points, the strong substantive improvements and the confirmed value recognized by reviewers, and recommend acceptance of our work.

---

### Meta-Review · Area_Chair_2Qm2 · 2026-01-07

**Summary:**

The paper introduces a large-scale, domain-specific benchmark for more rigorous evaluation of graph retrieval-augmented generation systems, with a focus on challenging questions that require , college-level knowledge and multi-hop reasoning.

The initial reviewers evaluated this paper as borderline, with two scores marginally above the acceptance threshold (6) and two marginally below the acceptance threshold (4). The reviewers generally agree that the paper is well written and introduces a potentially useful benchmark.

However, the paper raises several critical issues:

1. Limited domain scope and dataset size (e.g., only 1,018 questions across 16 topics within computer science). Reviewer Hctz raised a serious concern that the benchmark may not meet ICLR standards or appeal to a broader audience.
2. The absence of comparisons with other benchmark datasets.
3. The overall value of the benchmark remains unclear, as it is not sufficiently demonstrated whether performance on the proposed benchmark is generalizable, meaning that strong performance indicates that the model is suitable for some specific downstream applications.

**Reviewer Concerns:**

The authors improved the paper during the rebuttal. For instance, as suggested by a reviewer, the authors added DeepSeek-V3 and Qwen3-32B as LLM baselines. Originally, GPT-4o-mini was the only LLM baseline.

Despite the rebuttal, the major concerns below were not sufficiently addressed.

1. The authors discussed HippoRAG, which contains approximately 3,000 questions and is already larger than the benchmark dataset introduced in this paper. In addition, its follow-up work, HippoRAG2, includes an even larger set of questions (5,000+) drawn from seven different sources. This highlights that the proposed benchmark dataset has a smaller number of questions and a potentially narrower scope, as its topics are restricted to computer science.
2. The authors compared their benchmark only against HippoRAG, without broader comparisons to other recent benchmarks.
3. Including the value of the benchmark, several claims are supported primarily by qualitative discussion, without substantial empirical evidence or supporting data.

**Reviewer Scores:**

The authors introduce a new benchmark for evaluating GraphRAG systems, providing more challenging questions that require college-level knowledge and multi-hop reasoning, which are not well covered by prior RAG benchmarks, according to the authors’ claim. The authors also addressed some reviewer concerns during the rebuttal. However, multiple reviewers raised serious concerns about the limited scope confined to computer science, the relatively small dataset size, and the lack of clear advantages over more recent benchmarks. In addition, the generalizability and overall merit of the benchmark are not fully supported by the presented data. For these reasons, rejection is recommended.

---

### Decision · Program_Chairs · 2026-01-26

Reject